# Provable Benefit of Annealed Langevin Monte Carlo for Non-log-concave Sampling

**Wei Guo, Molei Tao, Yongxin Chen**
Georgia Institute of Technology
{wei.guo,mtao,yongchen}@gatech.edu

## Abstract

We consider the outstanding problem of sampling from an unnormalized density that may be non-log-concave and multimodal. To enhance the performance of simple Markov chain Monte Carlo (MCMC) methods, techniques of annealing type have been widely used. However, quantitative theoretical guarantees of these techniques are under-explored. This study takes a first step toward providing a non-asymptotic analysis of annealed MCMC. Specifically, we establish, for the first time, an oracle complexity of $\widetilde{O}\left(\frac{d\beta^2\mathcal{A}^2}{\varepsilon^6}\right)$ for the simple annealed Langevin Monte Carlo algorithm to achieve $\varepsilon^2$ accuracy in Kullback-Leibler divergence to the target distribution $\pi \propto \mathrm{e}^{-V}$ on $\mathbb{R}^d$ with $\beta$-smooth potential $V$. Here, $\mathcal{A}$ represents the action of a curve of probability measures interpolating the target distribution $\pi$ and a readily sampleable distribution.

## 1 Introduction

We study the task of efficient sampling from a probability distribution $\pi \propto \mathrm{e}^{-V}$ on $\mathbb{R}^d$. This fundamental problem is pivotal across various fields including computational statistics (Liu, 2004; Brooks et al., 2011), Bayesian inference (Gelman et al., 2013), statistical physics (Newman & Barkema, 1999), and finance (Dagpunar, 2007), and has been extensively studied in the literature (Chewi, 2024). The most common approach to this problem is Markov Chain Monte Carlo (MCMC), among which Langevin Monte Carlo (LMC) (Durmus et al., 2019; Vempala & Wibisono, 2019; Chewi et al., 2022; Mousavi-Hosseini et al., 2023) is a particularly popular choice. LMC can be understood as a time-discretization of a diffusion process, known as Langevin diffusion (LD), whose stationary distribution is the target distribution $\pi$, and has been attractive partly due to its robust performance despite conceptual simplicity.

Although LMC and its variants converge rapidly when the target distribution $\pi$ is strongly log-concave or satisfies isoperimetric inequalities such as the log-Sobolev inequality (LSI) (Durmus et al., 2019; Vempala & Wibisono, 2019; Chewi et al., 2022), its effectiveness diminishes when dealing with target distributions that are not strongly log-concave or are multimodal, such as mixtures of Gaussians. In such scenarios, the sampler often becomes confined to a single mode, severely limiting its ability to explore the entire distribution effectively. This results in significant challenges in transitioning between modes, which can dramatically increase the mixing time, making it exponential in problem parameters such as dimension, distance between modes, etc. (Dong & Tong, 2022; Ma et al., 2019). Such limitations highlight the need for enhanced MCMC methodologies that can efficiently navigate the complex landscapes of multimodal distributions, thereby improving convergence rates and overall sampling efficiency.

To address the challenges posed by multimodality, techniques around the notion of annealing have been widely employed (Gelfand et al., 1990; Neal, 2001). The general philosophy involves constructing a sequence of intermediate distributions $\pi_0, \pi_1, ..., \pi_M$ that bridge the gap between an easily samplable distribution $\pi_0$ (e.g., Gaussian or Dirac-like), and the target distribution $\pi_M = \pi$. The process starts with sampling from $\pi_0$ and progressively samples from each subsequent distribution until $\pi_M$ is reached. When $\pi_i$ and $\pi_{i+1}$ are close enough, approximate samples from $\pi_i$ can serve as a warm start for sampling from $\pi_{i+1}$, thereby facilitating this transition. Employing LMC within this framework gives rise to what is known as the annealed LMC algorithm, which is the focus of

our study. Despite its empirical success (Song & Ermon, 2019; 2020; Zilberstein et al., 2023; 2024), a thorough theoretical understanding of annealed LMC, particularly its non-asymptotic complexity bounds, remains elusive.

In this work, we take a first step toward developing a non-asymptotic analysis of annealed MCMC. Utilizing the Girsanov theorem to quantify the differences between the sampling dynamics and a reference process, we derive an upper bound on the error of annealed MCMC, which consists of two key terms. The first term is the ratio of an action functional in Wasserstein geometry, induced by optimal transport, to the duration of the process. This term decreases when the annealing schedule is sufficiently slow. The second term captures the discretization error inherent in practical implementations. Our approach challenges the traditional view that annealed MCMC is simply a series of warm starts. Analyses based on this perspective typically require assumptions of log-concavity or isoperimetric inequalities. In contrast, our theoretical framework for annealed MCMC dispenses with these assumptions, marking a significant shift in the understanding and analysis of annealed MCMC.

**Contributions.** Our key technical contributions are summarized as follows.

• We propose a novel strategy to analyze the non-asymptotic complexity bounds of annealed MCMC algorithms, bypassing the need for assumptions such as log-concavity or isoperimetry.

• In Sec. 4, we investigate the annealed LD, which involves running LD with a dynamically changing target distribution. We derive a notable bound on the time required to simulate the SDE for achieving $\varepsilon^2$-accuracy in KL divergence.

• Building on the insights from the analysis of the continuous dynamic and incorporating discretization errors, we establish a non-asymptotic oracle complexity bound for annealed LMC in Sec. 5, which is applicable to a wide range of annealing schemes.

**Comparison of quantitative results.** The quantitative results are summarized and compared to other sampling algorithms in Tab. 1. For algorithms requiring isoperimetric assumptions, we include Langevin Monte Carlo (LMC, Vempala & Wibisono (2019)) and Proximal Sampler (PS, Fan et al. (2023)), which converges rapidly but do not have theoretical guarantees without isoperimetric assumptions. For isoperimetry-free samplers, we include Simulated Tempering Langevin Monte Carlo (STLMC, Ge et al. (2018b)), a tempering-based sampler converging rapidly for a specific family of non-log-concave distributions, and three algorithms inspired by score-based generative models: Reverse Diffusion Monte Carlo (RDMC, Huang et al. (2024a)), Recursive Score Diffusion-based Monte Carlo (RS-DMC, Huang et al. (2024b)), and Zeroth Order Diffusion-Monte Carlo (ZOD-MC, He et al. (2024)), which involve simulating the time-reversal of the Ornstein-Uhlenbeck (OU) process and require estimating the scores of the intermediate distributions. Notably, our approach operates under the least stringent assumptions and exhibits the most favorable $\varepsilon$-dependence among all isoperimetry-free sampling methods.

**Related works.** We provide a brief overview of the literature, mainly focusing on the algorithms for non-log-concave sampling and their theoretical analysis.

**1.** Samplers based on tempering. The fundamental concept of tempering involves sampling the system at various temperatures simultaneously: at higher temparatures, the distribution flattens, allowing particles to easily transition between modes, while at lower temperatures, particles can more effectively explore local structures. In simulated tempering (Marinari & Parisi, 1992; Woodard et al., 2009), the system's temperature is randomly switched, while in parallel tempering (also known as replica exchange) (Swendsen & Wang, 1986; Lee & Shen, 2023), the temperatures of two particles are swapped according to a specific rule. However, quantitative theoretical results for tempering are limited, and the existing results (e.g., Ge et al. (2018a;b); Dong & Tong (2022)) apply primarily to certain special classes of non-log-concave distributions.

**2.** Samplers based on general diffusions. Inspired by score-based diffusion models (Ho et al., 2020; Song et al., 2021a;b), recent advances have introduced sampling methods that reverse the OU process, as detailed in Huang et al. (2024a;b); He et al. (2024). These samplers exhibit reduced sensitivity to isoperimetric conditions, but rely on estimating score functions (gradients of log-density)

Table 1: Comparison of oracle complexities in terms of $d, \varepsilon$, and the LSI constant for sampling from $\pi \propto \mathrm{e}^{-V}$. "poly$(\cdot)$" indicates a polynomial dependence on the specified parameters.

| Algorithm | Isoperimetric Assumptions | Other Assumptions | Criterion | Complexity |
|---|---|---|---|---|
| LMC | $C$-LSI | Potential smooth | $\varepsilon^2$, KL$(\cdot\|\pi)$ | $\widetilde{O}(C^2 d\varepsilon^{-2})$ |
| PS | $C$-LSI | Potential smooth | $\varepsilon$, TV | $\widetilde{O}(C d^{1/2} \log \varepsilon^{-1})$ |
| STLMC | / | Translated mixture of a well-conditioned distribution | $\varepsilon$, TV | $O(\mathrm{poly}(d, \varepsilon^{-1}))$ |
| RDMC | / | Potential smooth, nearly convex at $\infty$ | $\varepsilon$, TV | $O(\mathrm{poly}(d)\mathrm{e}^{\mathrm{poly}(\varepsilon^{-1})})$ |
| RS-DMC | / | Potential smooth | $\varepsilon^2$, KL$(\pi\|\cdot)$ | $\exp(O(\log^3 d\varepsilon^{-2}))$ |
| ZOD-MC | / | Potential growing at most quadratically | $\varepsilon$, TV + W$_2$ | $\exp(\widetilde{O}(d)O(\log \varepsilon^{-1}))$ |
| **ALMC (ours)** | / | Potential smooth | $\varepsilon^2$, KL$(\pi\|\cdot)$ | $\widetilde{O}(d\mathcal{A}(d)^2\varepsilon^{-6})$ |

via importance sampling, which poses significant challenges in high-dimensional settings. Concurrently, studies such as Zhang & Chen (2022); Vargas et al. (2023; 2024); Richter & Berner (2024) have employed neural networks to approximate unknown drift terms, enabling an SDE to transport a readily sampleable distribution to the target distribution. This approach has shown excellent performance in handling complex distributions, albeit at the expense of significant computational resources required for neural network training. In contrast, annealed LMC runs on a known interpolation of probability distributions, thus simplifying sampling by obviating the need for intensive score estimation or neural network training.

**3.** Non-asymptotic analysis for non-log-concave sampling. Drawing upon the stationary-point analysis in non-convex optimization, the seminal work Balasubramanian et al. (2022) characterizes the convergence of non-log-concave sampling via Fisher divergence. Subsequently, Cheng et al. (2023) applies this methodology to examine the local mixing of LMC. However, Fisher divergence is a relatively weak criterion compared to more commonly employed metrics such as total-variational distance or Wasserstein distances. In contrast, our study provides a convergence guarantee in terms of KL divergence, which implies convergence in total-variation distance and offers a stronger result.

**Notations and definitions.** For $a, b \in \mathbb{R}$, we define $[\![a, b]\!] := [a, b] \cap \mathbb{Z}$, $a \wedge b := \min(a, b)$, and $a \vee b := \max(a, b)$. For $a, b > 0$, the notations $a \lesssim b$, $a = O(b)$, and $b = \Omega(a)$ indicate that $a \leq Cb$ for some universal constant $C > 0$, and the notations $a \asymp b$ and $a = \Theta(b)$ stand for both $a = O(b)$ and $b = O(a)$. $\widetilde{O}(\cdot)$ hides logarithmic dependence in $O(\cdot)$. A function $U \in C^2(\mathbb{R}^d)$ is $\alpha(> 0)$-strongly-convex if $\nabla^2 U \succeq \alpha I$, and is $\beta(> 0)$-smooth[1] if $-\beta I \preceq \nabla^2 U \preceq \beta I$. The total-variation (TV) distance is defined as $\mathrm{TV}(\mu, \nu) = \sup_{A \subset \mathbb{R}^d} |\mu(A) - \nu(A)|$, and the Kullback-Leibler (KL) divergence is defined as $\mathrm{KL}(\mu\|\nu) = \mathbb{E}_\mu \log \frac{\mathrm{d}\mu}{\mathrm{d}\nu}$. $\|\cdot\|$ represents the $\ell^2$ norm on $\mathbb{R}^d$. For $f : \mathbb{R}^d \to \mathbb{R}^{d'}$ and a probability measure $\mu$ on $\mathbb{R}^d$, $\|f\|_{L^2(\mu)} := \left(\int \|f\|^2 \mathrm{d}\mu\right)^{\frac{1}{2}}$, and the second-order moment of $\mu$ is defined as $\mathbb{E}_\mu \|\cdot\|^2$.

## 2 PRELIMINARIES

### 2.1 STOCHASTIC DIFFERENTIAL EQUATIONS AND GIRSANOV THEOREM

A stochastic differential equation (SDE) $X = (X_t)_{t \in [0,T]}$ is a stochastic process on $\Omega = C([0, T]; \mathbb{R}^d)$, the space of continuous functions from $[0, T]$ to $\mathbb{R}^d$. The dynamics of $X$ are typically represented by the equation $\mathrm{d}X_t = b_t(X)\mathrm{d}t + \sigma_t(X)\mathrm{d}B_t$, $t \in [0, T]$, where $(B_t)_{t \in [0,T]}$ is a standard Brownian motion in $\mathbb{R}^d$, and $b_t(X) \in \mathbb{R}^d$, $\sigma_t(X) \in \mathbb{R}^{d \times d}$ depends on $(X_s)_{s \in [0,t]}$.

---

[1]Smoothness has several different definitions such as being $C^1$ or $C^\infty$, and the one used here is also a common one appearing in many standard textbooks in optimization and sampling, such as Chewi (2024).

The **path measure** of $X$, denoted $\mathbb{P}^X$, characterizes the distribution of $X$ over $\Omega$ and is defined by $\mathbb{P}^X(A) = \Pr(X \in A)$ for all measurable subset $A$ of $\Omega$. The following lemma, as a corollary of the Girsanov theorem (Üstünel & Zakai, 2013), provides a methodology for computing the KL divergence between two path measures and serves as a crucial technical tool in our proof.

**Lemma 1.** *Assume we have the following two SDEs on $\Omega$:*

$$\mathrm{d}X_t = a_t(X)\mathrm{d}t + \sqrt{2}\mathrm{d}B_t,\ X_0 \sim \mu; \qquad \mathrm{d}Y_t = b_t(Y)\mathrm{d}t + \sqrt{2}\mathrm{d}B_t,\ Y_0 \sim \nu.$$

*Let $\mathbb{P}^X$ and $\mathbb{P}^Y$ denote the path measures of $X$ and $Y$, respectively. Then*

$$\mathrm{KL}(\mathbb{P}^X \| \mathbb{P}^Y) = \mathrm{KL}(\mu \| \nu) + \frac{1}{4}\mathbb{E}_{X \sim \mathbb{P}^X} \int_0^T \|a_t(X) - b_t(X)\|^2 \mathrm{d}t.$$

## 2.2 LANGEVIN DIFFUSION AND LANGEVIN MONTE CARLO

The **Langevin diffusion (LD)** with target distribution $\pi \propto \mathrm{e}^{-V}$ is the solution to the SDE

$$\mathrm{d}X_t = -\nabla V(X_t)\mathrm{d}t + \sqrt{2}\mathrm{d}B_t,\ t \in [0, \infty);\ X_0 \sim \mu_0. \tag{1}$$

It is well-known that under mild conditions, $\pi$ is the unique stationary distribution this SDE, and when $\pi$ has good regularity properties, the marginal distribution of $X_t$ converges to $\pi$ as $t \to +\infty$, so we can sample from $\pi$ by simulating Eq. (1) for a long time. However, in most of the cases, LD is intractable to simulate exactly, and the Euler-Maruyama discretization of Eq. (1) leads to the **Langevin Monte Carlo (LMC)** algorithm. LMC with step size $h > 0$ and target distribution $\pi \propto \mathrm{e}^{-V}$ is a Markov chain $\{X_{kh}\}_{k=0,1,\ldots}$ constructed by iterating the following update rule:

$$X_{(k+1)h} = X_{kh} - h\nabla V(X_{kh}) + \sqrt{2}(B_{(k+1)h} - B_{kh}),\ k = 0, 1, \ldots;\ X_0 \sim \mu_0, \tag{2}$$

where $\{B_{(k+1)h} - B_{kh}\}_{k=0,1,\ldots} \overset{\mathrm{i.i.d.}}{\sim} \mathcal{N}(0, hI)$.

## 2.3 ISOPERIMETRIC INEQUALITIES

A probability measure $\pi$ on $\mathbb{R}^d$ satisfies a **log-Sobolev inequality (LSI)** with constant $C$, or $C$-LSI, if for all $f \in C^1(\mathbb{R}^d)$ with $\mathbb{E}_\pi f^2 > 0$,

$$\mathbb{E}_\pi f^2 \log \frac{f^2}{\mathbb{E}_\pi f^2} \leq 2C\mathbb{E}_\pi \|\nabla f\|^2.$$

A probability measure $\pi$ on $\mathbb{R}^d$ satisfies a **Poincaré inequality (PI)** with constant $C$, or $C$-PI, if for all $f \in C^1(\mathbb{R}^d)$,

$$\mathrm{Var}_\pi f \leq C\mathbb{E}_\pi \|\nabla f\|^2.$$

It is worth noting that $\alpha$-strongly-log-concave distributions satisfy $\frac{1}{\alpha}$-LSI, and $C$-LSI implies $C$-PI (Bakry et al., 2014). It is established in Vempala & Wibisono (2019) that when $\pi \propto \mathrm{e}^{-V}$ satisfies $C$-LSI, the LD converges exponentially fast in KL divergence; furthermore, when the potential $V$ is $\beta$-smooth, the LMC also converges exponentially with a bias that vanishes when the step size approaches 0.

## 2.4 WASSERSTEIN DISTANCE AND CURVES OF PROBABILITY MEASURES

We briefly introduce several fundamental concepts in optimal transport, and direct readers to authoritative textbooks (Villani, 2008; 2021; Ambrosio et al., 2008; 2021) for an in-depth exploration.

For two probability measures $\mu, \nu$ on $\mathbb{R}^d$ with finite second-order moments, the **Wasserstein-2 ($W_2$) distance** between $\mu$ and $\nu$ is defined as

$$W_2(\mu, \nu) = \inf_{\gamma \in \Pi(\mu,\nu)} \left( \int \|x - y\|^2 \gamma(\mathrm{d}x, \mathrm{d}y) \right)^{\frac{1}{2}},$$

where $\Pi(\mu, \nu)$ is the set of all couplings of $(\mu, \nu)$, i.e., probability measure $\gamma$ on $\mathbb{R}^d \times \mathbb{R}^d$ with $\gamma(A \times \mathbb{R}^d) = \mu(A)$ and $\gamma(\mathbb{R}^d \times A) = \nu(A)$, for all measurable set $A \subset \mathbb{R}^d$.

Given a vector field $v = (v_t : \mathbb{R}^d \to \mathbb{R}^d)_{t \in [a,b]}$ and a curve of probability measures $\rho = (\rho_t)_{t \in [a,b]}$ on $\mathbb{R}^d$ with finite second-order moments, we say that $v$ **generates** $\rho$ if the continuity equation $\partial_t \rho_t + \nabla \cdot (\rho_t v_t) = 0$, $t \in [a,b]$ holds. The **metric derivative** of $\rho$ at $t \in [a,b]$ is defined as

$$|\dot{\rho}|_t := \lim_{\delta \to 0} \frac{W_2(\rho_{t+\delta}, \rho_t)}{|\delta|},$$

which can be interpreted as the "speed" of this curve. If $|\dot{\rho}|_t$ exists and is finite for all $t \in [a,b]$, we say that $\rho$ is **absolutely continuous (AC)**. AC is a fairly weak regularity condition on the curve of probability measures. In Lem. 4, we present a sufficient condition for AC as a formal statement.

The metric derivative and the continuity equation are closely related by the following important fact from Ambrosio et al. (2008, Theorems 8.3.1 and 8.4.5):

**Lemma 2.** *For an AC curve of probability measures $(\rho_t)_{t \in [a,b]}$, any vector field $(v_t)_{t \in [a,b]}$ that generates $(\rho_t)_{t \in [a,b]}$ satisfies $|\dot{\rho}|_t \leq \|v_t\|_{L^2(\rho_t)}$ for a.e. $t \in [a,b]$. Moreover, there exists a unique vector field $(v_t^*)_{t \in [a,b]}$ generating $(\rho_t)_{t \in [a,b]}$ that satisfies $|\dot{\rho}|_t = \|v_t^*\|_{L^2(\rho_t)}$ for a.e. $t \in [a,b]$.*

Finally, we define the **action** of an AC curve of probability measures $(\rho_t)_{t \in [a,b]}$ as $\int_a^b |\dot{\rho}|_t^2 \mathrm{d}t$. As will be shown in the next section, the action is a key property characterizing the effectiveness of a curve in annealed sampling. The following lemma provides additional intuition about the action and may be helpful for interested readers.

**Lemma 3.** *Given an AC curve of probability measures $(\rho_t)_{t \in [0,1]}$, and let $\mathcal{A}$ be its action. Then*

**1.** $\mathcal{A} \geq W_2^2(\rho_0, \rho_1)$, *and the equality is attained when $(\rho_t)_{t \in [0,1]}$ is a constant-speed Wasserstein geodesic, i.e., let two random variables $(X_0, X_1)$ follow the optimal coupling of $(\rho_0, \rho_1)$ such that $\mathbb{E}\|X_0 - X_1\|^2 = W_2^2(\rho_0, \rho_1)$, and define $\rho_t$ as the law of $(1-t)X_0 + tX_1$.*

**2.** *If $\rho_t$ satisfies $C_{\mathrm{LSI}}(\rho_t)$-LSI for all $t$, then $\mathcal{A} \leq \int_0^1 C_{\mathrm{LSI}}(\rho_t)^2 \|\partial_t \nabla \log \rho_t\|_{L^2(\rho_t)}^2 \mathrm{d}t$.*

**3.** *If $\rho_t$ satisfies $C_{\mathrm{PI}}(\rho_t)$-PI for all $t$, then $\mathcal{A} \leq \int_0^1 2C_{\mathrm{PI}}(\rho_t) \|\partial_t \log \rho_t\|_{L^2(\rho_t)}^2 \mathrm{d}t$.*

The proof of Lem. 3 can be found in App. A. The lower bound above can be derived by definition or via variational representations of action and $W_2$ distance, while the two upper bounds relate the action to the isoperimetric inequalities, indicating that finite action is a weaker assumption than requiring the LSI or PI constants along the trajectory. In fact, these bounds may not be tight, as demonstrated by the following surprising example, in which the action is polynomial with respect to certain problem parameters and even independent of some of them, whereas the LSI and PI constants exhibit exponential dependence. Its proof is detailed in App. A.

**Example 1.** *Consider the curve that bridges a target distribution $\rho_0$ to a readily sampleable distribution $\rho_1$, in the form $\rho_t = \rho_0 * \mathcal{N}(0, 2StI)$, $t \in [0,1]$, for some large positive number $S$. Let $\mathcal{A}$ be the action of this curve. Then*

**1.** $\mathcal{A} = S \int_0^S \|\nabla \log p_s\|_{L^2(p_s)}^2 \mathrm{d}s$, *where $p_s = \rho_{s/S} = \rho_0 * \mathcal{N}(0, 2sI)$, $s \in [0, S]$.*

**2.** *In particular, when $\rho_0$ is a mixture of Gaussian distribution $\sum_{i=1}^N w_i \mathcal{N}(\mu_i, \sigma^2 I)$, then $\mathcal{A} \leq \frac{Sd}{2} \log\left(1 + \frac{2S}{\sigma^2}\right)$ is independent of the number of components $N$, the weights $w_i$, and the means $\mu_i$. However, in general, the LSI and PI constants may depend exponentially on $\max_{i,j} \|\mu_i - \mu_j\|$, the maximum distance between the means.*

## 3 PROBLEM SETUP

Recall that the rationale behind annealing involves a gradual transition from $\pi_0$, a simple distribution that is easy to sample from, to $\pi_1 = \pi$, the more complex target distribution. Throughout this paper, we define a curve of probability measures $(\pi_\theta)_{\theta \in [0,1]}$, along which we will apply annealing-based MCMC for sampling from $\pi$. For now, we do not specify the exact form of this curve, but instead introduce the following mild regularity assumption:

**Assumption 1.** *Each $\pi_\theta$ has a finite second-order moment, and the curve $(\pi_\theta)_{\theta \in [0,1]}$ is AC with finite action $\mathcal{A} = \int_0^1 |\dot{\pi}|_\theta^2 \mathrm{d}\theta$.*

For the purpose of non-asymptotic analysis, we further introduce the following mild assumption on the target distribution $\pi \propto \mathrm{e}^{-V}$ on $\mathbb{R}^d$, which is widely used in the field of sampling (see Chewi (2024) for an overview):

**Assumption 2.** *The potential $V$ is $\beta$-smooth, and there exists a global minimizer $x_*$ of $V$ such that $\|x_*\| \leq R$. Moreover, $\pi$ has finite second-order moment.*

With this foundational setup, we now proceed to introduce the annealed LD and annealed LMC algorithms. Our goal is to characterize the non-asymptotic complexity of annealed MCMC algorithms for sampling from possibly non-log-concave distributions.

## 4 ANALYSIS OF ANNEALED LANGEVIN DIFFUSION

To elucidate the concept of annealing more clearly, we first consider the **annealed Langevin diffusion (ALD)** algorithm, which samples from the $\pi \propto \mathrm{e}^{-V}$ by running LD with a dynamically changing target distribution. For the sake of this discussion, we assume the following: (i) we can exactly sample from $\pi_0$, (ii) the scores $(\nabla \log \pi_\theta)_{\theta \in [0,1]}$ are known in closed form, and (iii) we can exactly simulate any SDE with known drift and diffusion terms.

Fix a sufficiently long time $T$. We define a reparametrized curve of probability measures $(\widetilde{\pi}_t := \pi_{t/T})_{t \in [0,T]}$. Starting with an initial sample $X_0 \sim \pi_0 = \widetilde{\pi}_0$, we run the SDE

$$\mathrm{d}X_t = \nabla \log \widetilde{\pi}_t(X_t)\mathrm{d}t + \sqrt{2}\mathrm{d}B_t, \ t \in [0,T], \tag{3}$$

and ultimately output $X_T \sim \nu^{\mathrm{ALD}}$ as an approximate sample from the target distribution $\pi$. Intuitively, when $\widetilde{\pi}_t$ is changing slowly, the distribution of $X_t$ should closely resemble $\widetilde{\pi}_t$, leading to an output distribution $\nu^{\mathrm{ALD}}$ that approximates the target distribution. This turns out to be true, as is confirmed by the following theorem, which provides a convergence guarantee for the ALD process.

**Theorem 1.** *When choosing $T = \frac{\mathcal{A}}{4\varepsilon^2}$, it follows that $\mathrm{KL}(\pi \| \nu^{\mathrm{ALD}}) \leq \varepsilon^2$.*

*Proof.* Let $\mathbb{Q}$ be the path measure of ALD (Eq. (3)) initialized at $X_0 \sim \widetilde{\pi}_0$, and define $\mathbb{P}$ as the path measure corresponding to the following reference SDE:

$$\mathrm{d}X_t = (\nabla \log \widetilde{\pi}_t + v_t)(X_t)\mathrm{d}t + \sqrt{2}\mathrm{d}B_t, \ X_0 \sim \widetilde{\pi}_0, \ t \in [0,T]. \tag{4}$$

The vector field $v = (v_t)_{t \in [0,T]}$ is designed such that $X_t \sim \widetilde{\pi}_t$ for all $t \in [0,T]$. According to the Fokker-Planck equation[2], this is equivalent to the following PDE:

$$\partial_t \widetilde{\pi}_t = -\nabla \cdot (\widetilde{\pi}_t (\nabla \log \widetilde{\pi}_t + v_t)) + \Delta \widetilde{\pi}_t = -\nabla \cdot (\widetilde{\pi}_t v_t), \ t \in [0,T],$$

which means that $v$ generates $\left(\widetilde{\pi}_t := \pi_{t/T}\right)_{t \in [0,T]}$. We can compute $\mathrm{KL}(\mathbb{P}\|\mathbb{Q})$ using Lem. 1:

$$\mathrm{KL}(\mathbb{P}\|\mathbb{Q}) = \frac{1}{4}\mathbb{E}_{\mathbb{P}}\int_0^T \|v_t(X_t)\|^2\mathrm{d}t = \frac{1}{4}\int_0^T \|v_t\|_{L^2(\widetilde{\pi}_t)}^2\mathrm{d}t.$$

Leveraging Lem. 2, among all vector fields $v$ that generate $\left(\widetilde{\pi}_t := \pi_{t/T}\right)_{t \in [0,T]}$, we can choose the one that minimizes $\|v_t\|_{L^2(\widetilde{\pi}_t)}$, thereby making $\|v_t\|_{L^2(\widetilde{\pi}_t)} = |\dot{\widetilde{\pi}}|_t$, the metric derivative. With the reparameterization $\widetilde{\pi}_t = \pi_{t/T}$, we have the following relation by chain rule:

$$|\dot{\widetilde{\pi}}|_t = \lim_{\delta \to 0}\frac{W_2(\widetilde{\pi}_{t+\delta}, \widetilde{\pi}_t)}{|\delta|} = \lim_{\delta \to 0}\frac{W_2(\pi_{(t+\delta)/T}, \pi_{t/T})}{T|\delta/T|} = \frac{1}{T}|\dot{\pi}|_{t/T}.$$

Employing the change-of-variable formula leads to

$$\mathrm{KL}(\mathbb{P}\|\mathbb{Q}) = \frac{1}{4}\int_0^T |\dot{\widetilde{\pi}}|_t^2\mathrm{d}t = \frac{1}{4T}\int_0^1 |\dot{\pi}|_\theta^2\mathrm{d}\theta = \frac{\mathcal{A}}{4T}.$$

---

[2]Here, we assume that the solution (in terms of the curve of probability measures) of the Fokker-Planck equation given the drift and diffusion terms exists and is unique, which holds under weak regularity conditions (Le Bris & Lions, 2008).

Finally, using data-processing inequality (see, e.g., Chewi (2024, Theorem 1.5.3)), with $T = \frac{\mathcal{A}}{4\varepsilon^2}$,

$$\mathrm{KL}(\pi \| \nu^{\mathrm{ALD}}) = \mathrm{KL}(\mathbb{P}_T \| \mathbb{Q}_T) \leq \mathrm{KL}(\mathbb{P} \| \mathbb{Q}) = \varepsilon^2,$$

where $\mathbb{P}_T$ and $\mathbb{Q}_T$ stand for the marginal distributions of $\mathbb{P}$ and $\mathbb{Q}$ at time $T$, respectively. $\qquad\square$

Let us delve deeper into the mechanics of the ALD. Although at time $t$ the SDE (Eq. (3)) targets the distribution $\widetilde{\pi}_t$, the distribution of $X_t$ does not precisely align with $\widetilde{\pi}_t$. Nevertheless, by choosing a sufficiently long time $T$, we actually move on the curve $(\pi_\theta)_{\theta \in [0,1]}$ sufficiently slowly, thus minimizing the discrepancy between path measure of $(X_t)_{t \in [0,T]}$ and the reference curve $(\widetilde{\pi}_t)_{t \in [0,T]}$. Using the data-processing inequality, we can upper bound the error between the marginal distributions at time $T$ by the joint distributions of the two paths. In essence, *moving more slowly, sampling more precisely*.

Our analysis primarily addresses the *global* error across the entire curve of probability measures, rather than focusing solely on the *local* error at time $T$. This approach is inspired by Dalalyan & Tsybakov (2012); Chen et al. (2023), and stands in contrast to the isoperimetry-based analyses of LD (e.g., Vempala & Wibisono (2019); Chewi et al. (2022); Balasubramanian et al. (2022)), which focus on the decay of the KL divergence from the distribution of $X_t$ to the target distribution, and require LSI to bound the time derivative of this quantity. Notably, the total time $T$ needed to run the SDE depends solely on the action of the curve $(\pi_\theta)_{\theta \in [0,1]}$, obviating the need for assumptions related to log-concavity or isoperimetry.

It is also worth noting that ALD plays a critical role in the field of non-equilibrium stochastic thermodynamics (Seifert, 2012). Recently, a refinement of the fluctuation theorem was discovered in Chen et al. (2020); Fu et al. (2021), showing that the irreversible entropy production in a stochastic thermodynamic system is equal to the ratio of a similar action integral and the duration of the process $T$, closely resembling Thm. 1.

## 5 ANALYSIS OF ANNEALED LANGEVIN MONTE CARLO

It is crucial to recognize that, in practice, running the algorithm requires knowledge of the score functions $(\nabla \log \pi_\theta)_{\theta \in [0,1]}$ in closed form. Additionally, simulating ALD (Eq. (3)) necessitates discretization schemes, which introduces further errors. This section presents a detailed non-asymptotic convergence analysis for the **annealed Langevin Monte Carlo (ALMC)** algorithm, which is a practical approach for real-world implementations.

The outline of this section is as follows. We will first introduce a family of the interpolation curve $(\pi_\theta)_{\theta \in [0,1]}$ that serves as the basis for our analysis. Next, we propose a discretization scheme tailored to the special structure of $(\pi_\theta)_{\theta \in [0,1]}$. Finally, we state the complexity analysis in Thm. 2, and provide an example demonstrating the improvement of our bounds over existing results.

### 5.1 CHOICE OF THE INTERPOLATION CURVE

We consider the following curve of probability measures on $\mathbb{R}^d$:

$$\pi_\theta \propto \exp\left(-\eta(\theta)V - \frac{\lambda(\theta)}{2}\|\cdot\|^2\right), \ \theta \in [0,1], \tag{5}$$

where the functions $\eta(\cdot)$ and $\lambda(\cdot)$ are called the **annealing schedule**. These functions must be differentiable and monotonic, satisfying the boundary conditions $\eta_0 = \eta(0) \nearrow \eta(1) = 1$ and $\lambda_0 = \lambda(0) \searrow \lambda(1) = 0$. The values of $\eta_0 \in [0,1]$ and $\lambda_0 \in (1, +\infty)$ are chosen so that $\pi_0$ corresponds a Gaussian distribution $\mathcal{N}\left(0, \lambda_0^{-1}I\right)$ or can be efficiently sampled within $\widetilde{O}(1)$ steps of rejection sampling (see Lem. 5 for details). We also remark that, under Assump. 2, $\pi_\theta$ has a finite second-order moment (see Lem. 8 the proof), ensuring the $\mathrm{W}_2$ distance between $\pi_\theta$'s is well-defined.

This flexible interpolation scheme includes many general cases. For example, Brosse et al. (2018) and Ge et al. (2020) used the schedule $\eta(\cdot) \equiv 1$, while Neal (2001) used the schedule $\lambda(\theta) = c(1 - \eta(\theta))$. The key motivation for this interpolation is that when $\theta = 0$, the quadratic term predominates, making the potential of $\pi_0$ strongly-convex with a moderate condition number, thus $\pi_0$ is easy to sample from; on the other hand, when $\theta = 1$, $\pi_1$ is just the target distribution $\pi$. As

$\theta$ gradually increases from 0 to 1, the readily sampleable distribution $\pi_0$ slowly transforms into the target distribution $\pi_1$.

## 5.2 THE DISCRETIZATION ALGORITHM

With the interpolation curve $(\pi_\theta)_{\theta \in [0,1]}$ chosen as above, a straightforward yet non-optimal method to discretize Eq. (3) involves employing the Euler-Maruyama scheme, i.e.,

$$X_{t+\Delta t} \approx X_t + \nabla \log \widetilde{\pi}_t(X_t)\Delta t + \sqrt{2}(B_{t+\Delta t} - B_t),\ 0 \le t < t + \Delta t \le T.$$

However, considering that $\nabla \log \widetilde{\pi}_t(x) = -\eta\left(\frac{t}{T}\right)\nabla V(x) - \lambda\left(\frac{t}{T}\right)x$, the integral of the linear term can be computed in closed form, so we can use the exponential-integrator scheme (Zhang & Chen, 2023; Zhang et al., 2023) to further reduce the discretization error. Given the total time $T$, we define a sequence of points $0 = \theta_0 < \theta_1 < ... < \theta_M = 1$, and set $T_\ell = T\theta_\ell$, $h_\ell = T(\theta_\ell - \theta_{\ell-1})$. The exponential-integrator scheme is then expressed as

$$\mathrm{d}X_t = \left(-\eta\left(\frac{t}{T}\right)\nabla V(X_{t_-}) - \lambda\left(\frac{t}{T}\right)X_t\right)\mathrm{d}t + \sqrt{2}\mathrm{d}B_t,\ t \in [0,T],\ X_0 \sim \pi_0, \quad (6)$$

where $t_- := T_{\ell-1}$ when $t \in [T_{\ell-1}, T_\ell)$, $\ell \in [\![1, M]\!]$. The explicit update rule is detailed in Alg. 1, with $x_\ell$ denoting $X_{T_\ell}$, and the derivation of Eq. (7) is presented in App. C.1. Notably, replacing $\nabla V(X_{t_-})$ with $\nabla V(X_t)$ recovers the ALD (Eq. (3)), and setting $\eta \equiv 1$ and $\lambda \equiv 0$ reduces to the classical LMC iterations.

---

**Algorithm 1:** Annealed LMC Algorithm

**Input:** Target distribution $\pi \propto \mathrm{e}^{-V}$, total time $T$, annealing schedule $\eta(\cdot)$ and $\lambda(\cdot)$, discrete points $\theta_0, ..., \theta_M$.

1 For $0 \le \theta < \theta' \le 1$, define $\Lambda_0(\theta', \theta) = \exp\left(-T\int_\theta^{\theta'} \lambda(u)\mathrm{d}u\right)$,
  $H(\theta', \theta) = T\int_\theta^{\theta'} \eta(u)\Lambda_0(u, \theta')\mathrm{d}u$, and $\Lambda_1(\theta', \theta) = \sqrt{2T\int_\theta^{\theta'} \Lambda_0^2(u, \theta')\mathrm{d}u}$;

2 Obtain a sample $x_0 \sim \pi_0$ (e.g., using rejection sampling (Alg. 2));

3 **for** $\ell = 1, 2, ..., M$ **do**

4     Sample an independent Gaussian noise $\xi_\ell \sim \mathcal{N}(0, I)$;

5     Update
$$x_\ell = \Lambda_0(\theta_\ell, \theta_{\ell-1})x_{\ell-1} - H(\theta_\ell, \theta_{\ell-1})\nabla V(x_{\ell-1}) + \Lambda_1(\theta_\ell, \theta_{\ell-1})\xi_\ell. \quad (7)$$

6 **end**

**Output:** $x_M \sim \nu^{\mathrm{ALMC}}$, an approximate sample from $\pi$.

---

We illustrate the ALMC algorithm in Fig. 1. The underlying intuition behind this algorithm is that by setting a sufficiently long total time $T$, the trajectory of the continuous dynamic (i.e., annealed LD) approaches the reference trajectory closely, as established in Thm. 1. Additionally, by adopting sufficiently small step sizes $h_1, ..., h_M$, the discretization error can be substantially reduced. Unlike traditional annealed LMC methods, which require multiple LMC update steps for each intermediate distribution $\pi_1, ..., \pi_M$, our approach views the annealed LMC as a discretization of a continuous-time process. Consequently, it is adequate to perform *a single step* of LMC towards each intermediate distribution $\widetilde{\pi}_{T_1}, ..., \widetilde{\pi}_{T_M}$, provided that $T$ is sufficiently large and the step sizes are appropriately small.

## 5.3 CONVERGENCE ANALYSIS OF THE ALGORITHM

The subsequent theorem provides a convergence guarantee for the annealed LMC algorithm, with a detailed proof available in App. C.2.

**Theorem 2.** *Under Assumps. 1 and 2, Alg. 1 can generate a distribution $\nu^{\mathrm{ALMC}}$ satisfying* $\mathrm{KL}(\pi\|\nu^{\mathrm{ALMC}}) \le \varepsilon^2$ *within*

$$\widetilde{O}\left(\frac{d\beta^2\mathcal{A}^2}{\varepsilon^6}\right)$$

*calls to the oracle of $V$ and $\nabla V$ in expectation.*

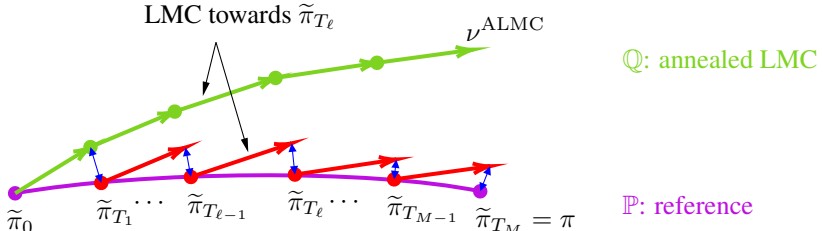

Figure 1: Illustration of the ALMC algorithm. The $\ell$-th green arrow, proceeding from left to right, represents one step of LMC towards $\widetilde{\pi}_{T_\ell}$ with step size $h_\ell$, while each red arrow corresponds to the application of the same transition kernel, initialized at $\widetilde{\pi}_{T_{\ell-1}}$ on the reference trajectory $\mathbb{P}$, which is depicted in purple. To evaluate $\mathrm{KL}(\mathbb{P}\|\mathbb{Q})$, the Girsanov theorem implies that we only need to bound the aggregate KL divergence across each small interval (i.e., the sum of the blue "distances").

*Sketch of Proof.* Let $\mathbb{Q}$ be the path measure of ALMC (Eq. (3)), the time-discretized sampling process, whose marginal distribution at time $T$ is the output distribution $\nu^{\mathrm{ALMC}}$. Again, let $\mathbb{P}$ denote the reference path measure of Eq. (4) used in the proof of Thm. 1, in which the same vector field $(v_t)_{t\in[0,T]}$ ensures that $X_t \sim \widetilde{\pi}_t$ under $\mathbb{P}$ for all $t \in [0,T]$. Applying Girsanov theorem (Lem. 1) and carefully dealing with the discretization error, we can upper bound $\mathrm{KL}(\mathbb{P}\|\mathbb{Q})$ by

$$\mathrm{KL}(\mathbb{P}\|\mathbb{Q}) \lesssim \sum_{\ell=1}^{M} \left( \frac{1 + \eta(\theta_\ell)^2 \beta^2 h_\ell^2}{T} \int_{\theta_{\ell-1}}^{\theta_\ell} |\dot{\pi}|_\theta^2 \mathrm{d}\theta + \eta(\theta_\ell)^2 \beta^2 dh_\ell^2 \left(1 + h_\ell \left(\beta\eta(\theta_\ell) + \lambda(\theta_{\ell-1})\right)\right) \right).$$

The first summation is governed by the total time $T$, which pertains to the convergence of the continuous dynamic (i.e., ALD). Setting $T \asymp \frac{A}{\varepsilon^2}$ ensures that the first summation remains $O(\varepsilon^2)$, provided that the step size $h_\ell$ is sufficiently small. The second summation addresses the discretization error, and it suffices to determine the appropriate value of $\theta_\ell$ to minimize $M$, the total number of calls to the oracle of $\nabla V$ for discretizing the SDE. Combining $M$ with the complexity of sampling from $\pi_0$ determines the overall complexity of the algorithm. $\qquad\square$

Once again, our analysis relies on bounding the global error between two path measures by Girsanov theorem. The annealing schedule $\eta(\cdot)$ and $\lambda(\cdot)$ influences the complexity exclusively through the action $\mathcal{A}$. This crucial identity, based on the choice of interpolation curve $(\pi_\theta)_{\theta\in[0,1]}$, significantly affects the effectiveness of ALMC. Our specific choice (Eq. (5)), among all interpolation curves with closed-form scores, aims at simplifying the discretization error analysis. However, alternative annealing schemes could be explored to find the optimal one in practice. For instance, a recent paper (Fan et al., 2024) proposed an annealing scheme $\pi_\theta(x) \propto \pi_0((1-a\theta)x)^{1-\theta}\pi_1\left(\frac{x}{b+(1-b)\theta}\right)^\theta$ for some $a \in [0,1]$ and $b \in (0,1]$. The extra scaling factor $\frac{1}{b+(1-b)\theta}$ might improve mixing speed. We leave the choice of the interpolation curve as a topic for future research.

We also note that our Assump. 2 encompasses strongly-log-concave target distributions. For sampling from these well-conditioned distributions via LMC, the complexity required to achieve $\varepsilon$-accuracy in TV distance scales as $\widetilde{O}(\varepsilon^{-2})$ (Chewi, 2024). However, using Pinsker inequality $\mathrm{KL} \geq 2\,\mathrm{TV}^2$, our complexity to meet the same error criterion is $O(\varepsilon^{-6})$, indicating a significantly higher computational demand. While our discretization error is as sharp as existing works, the main reason for our worse $\varepsilon$-dependence is due to the time required for the continuous dynamics to converge, which is based on Girsanov theorem rather than LSI. While sharpening the $\varepsilon$-dependence based on Girsanov theorem remains a challenge, our $O(\varepsilon^{-6})$ dependence still outperforms all the other bounds in Tab. 1 without isoperimetry assumptions.

Finally, we conclude by demonstrating an example of a class of mixture of Gaussian distributions, which illustrates how our analysis can improve the complexity bound from exponential to polynomial. The detailed proof is provided in App. D.

**Example 2.** *Consider a mixture of Gaussian distributions in $\mathbb{R}^d$ defined by $\pi = \sum_{i=1}^{N} p_i \mathcal{N}\left(y_i, \beta^{-1}I\right)$, where the weights $p_i > 0$, $\sum_{i=1}^{N} p_i = 1$, and $\|y_i\| = r$ for all $i \in [\![1, N]\!]$. Consequently, the potential $V = -\log \pi$ is $B$-smooth, where $B = \beta(4r^2\beta + 1)$. With an annealing*

*schedule defined by $\eta(\cdot) \equiv 1$ and $\lambda(\theta) = dB(1-\theta)^\gamma$ for some $1 \le \gamma = O(1)$, it follows that $\mathcal{A} = O\left(d(r^2\beta + 1)\left(r^2 + \frac{d}{\beta}\right)\right)$.*

To show the superiority of our theoretical results, consider a simplified scenario with $N = 2$, $y_1 = -y_2$, and $r^2 \gg \beta^{-1}$. By applying Ex. 2 and the Pinsker inequality, the total complexity required to obtain an $\varepsilon$-accurate sample in TV distance is $\widetilde{O}(d^3\beta^2 r^4(r^4\beta^2 \vee d^2)\varepsilon^{-6})$. In contrast, studies such as Schlichting (2019); Chen et al. (2021); Dong & Tong (2022) indicate that the LSI constant of $\pi$ is $\Omega(e^{\Theta(\beta r^2)})$, so existing bounds in Tab. 1 suggest that LMC, to achieve the same accuracy, can only provide an exponential complexity guarantee of $\widetilde{O}(e^{\Theta(\beta r^2)}d\varepsilon^{-2})$.

## 6 EXPERIMENTS

We conduct simple numerical experiments to verify the findings in Ex. 2, which demonstrate that for a certain class of mixture of Gaussian distributions, ALMC achieves polynomial convergence with respect to $r$. Specifically, we consider a target distribution comprising a mixture of 6 Gaussians with uniform weights $\frac{1}{6}$, means $\left\{\left(r\cos\frac{k\pi}{3}, r\sin\frac{k\pi}{3}\right) : k \in [\![0,5]\!]\right\}$, and the same covariance $0.1I$ (corresponding to $\beta = 10$ in Ex. 2). We experimented $r \in \{2, 5, 10, 15, 20, 25, 30\}$, and compute the number of iterations required for the empirical KL divergence from the target distribution to the sampled distribution to fall below $0.2$ (blue curve) and $0.1$ (orange curve). The results, displayed in Fig. 2, use a log-log scale for both axes. The near-linear behavior of the curves in this plot confirms that the complexity depends polynomially on $r$, validating our theory. Further details of the experimental setup and implementation can be found in App. F.

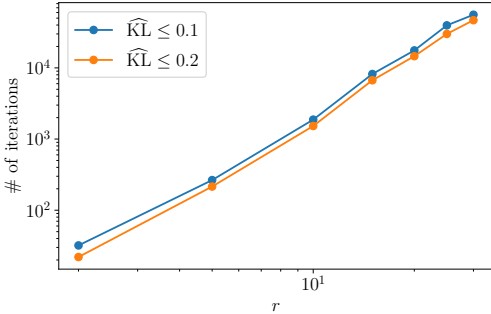

Figure 2: Relationship between norm of means $r$ and the number of iterations to reach $0.2$ and $0.1$ accuracy in KL divergence, both in $\log$ scale.

## 7 CONCLUSIONS AND FUTURE WORK

In this paper, we have explored the complexity of ALMC for sampling from a non-log-concave probability measure, circumventing the reliance on log-concavity or isoperimetric inequalities. Central to our analysis are the Girsanov theorem and optimal transport techniques, providing a novel approach. While our proof primarily focuses on the annealing scheme as described in Eq. (5), it can potentially be adapted to more general interpolations. Further exploration of these applications will be a key direction for future research. Technically, our proof methodology could be expanded to a broader range of target distributions beyond those with smooth potentials, such as those with Hölder-continuous gradients (Chatterji et al., 2020; Liang & Chen, 2022; Fan et al., 2023). Eliminating the assumption of global minimizers of the potential function would further enhance the practical applicability of our algorithm. Finally, while this work emphasizes the upper bounds for non-log-concave sampling, exploring the tightness of these bounds and investigating potential lower bounds for this problem (Chewi et al., 2023a;b) are intriguing avenues for future research.

ACKNOWLEDGMENTS

We would like to thank Ye He, Sinho Chewi, Matthew S. Zhang, Omar Chehab, Adrien Vacher, and Anna Korba for insightful discussions. MT is supported by the National Science Foundation under Award No. DMS-1847802, and WG and YC are supported by the National Science Foundation under Award No. ECCS-1942523 and DMS-2206576.

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

## A  PROOF OF RESULTS ON ACTION

**Lemma 4** (Sufficient Condition for Absolute Continuity (Informal)). *Assume that a curve of probability distributions $(\pi_\theta)_{\theta \in [0,1]}$ on $\mathbb{R}^d$ has a density $\pi : [0,1] \times \mathbb{R}^d \ni (\theta, x) \mapsto \pi_\theta(x) > 0$ that is jointly $C^1$. Then, this curve is AC.*

*Proof.* The following proof is informal. We leave the task of formalizing this statement for future work.

We first prove the result in one-dimension. Let $F_\theta(x) = \int_{-\infty}^x \pi_\theta(u) \mathrm{d}u$ be the c.d.f. of $\pi_\theta$, which is strictly increasing as $\pi_\theta(x) > 0$ everywhere. As a result, its inverse $F_\theta^{-1}$ is well-defined, and $F_\cdot^{-1}(\cdot)$ is also jointly $C^1$. We know from Ambrosio et al. (2008, Theorem 6.0.2) that

$$W_2^2(\pi_\theta, \pi_{\theta+\delta}) = \int_0^1 |F_{\theta+\delta}^{-1}(q) - F_\theta^{-1}(q)|^2 \mathrm{d}q.$$

The above quantity should be $O(\delta^2)$ as $\delta \to 0$ due to the $C^1$ property, and hence the curve is AC.

In $d$-dimensions, consider the one-dimensional subspace robust $W_2$ distance (Paty & Cuturi, 2019) defined as

$$S_2(\mu, \nu) = \sup_{v \in \mathbb{S}^{d-1}} W_2(v_\sharp \mu, v_\sharp \nu),$$

where $\mathbb{S}^{d-1} = \{v \in \mathbb{R}^d : \|v\| = 1\}$ is the set of unit vectors in $\mathbb{R}^d$, and $v_\sharp \mu$ (respectively, $v_\sharp \nu$) is the law of $\langle v, X \rangle$ when $X \sim \mu$ (respectively, $X \sim \nu$), which is a probability measure in $\mathbb{R}$. By Paty & Cuturi (2019, Proposition 2), $W_2(\mu, \nu) \leq \sqrt{d} S_2(\mu, \nu)$. Similar argument as above shows that $W_2^2(v_\sharp \pi_\theta, v_\sharp \pi_{\theta+\delta}) = O(\delta^2)$, and hence $W_2^2(\pi_\theta, \pi_{\theta+\delta}) = O(\delta^2)$. $\qquad \square$

**Proof of Lem. 3.**

**1.** By Cauchy-Schwarz inequality,

$$\mathcal{A} = \int_0^1 |\dot{\rho}|_t^2 \mathrm{d}t \int_0^1 1 \mathrm{d}t \geq \left( \int_0^1 |\dot{\rho}|_t \mathrm{d}t \right)^2.$$

It remains to prove that $W_2(\rho_0, \rho_1) \leq \int_0^1 |\dot{\rho}|_t \mathrm{d}t$. In fact, take any sequence $0 = t_0 < t_1 < ... < t_N = 1$, and let $\Delta t := \max_{1 \leq i \leq N}(t_i - t_{i-1})$. By triangle inequality of the $W_2$ distance,

$$
\begin{aligned}
W_2(\rho_0, \rho_1) &\leq \sum_{i=1}^N W_2(\rho_{t_i}, \rho_{t_{i-1}}) \\
&= \sum_{i=1}^N \frac{W_2(\rho_{t_i}, \rho_{t_{i-1}})}{t_i - t_{i-1}} (t_i - t_{i-1}) \\
&= \sum_{i=1}^N (|\dot{\rho}|_{t_i} + o(\Delta t))(t_i - t_{i-1}) \to \int_0^1 |\dot{\rho}|_t \mathrm{d}t.
\end{aligned}
$$

Another derivation is by investigating the variational representations of these two quantities. First, by Lem. 2, one can write

$$\mathcal{A} = \int_0^1 |\dot{\rho}|_t^2 \mathrm{d}t = \inf_{v_t: \, \partial_t \rho_t + \nabla \cdot (\rho_t v_t) = 0} \int_0^1 \|v_t\|_{L^2(\rho_t)}^2 \mathrm{d}t.$$

On the other hand, the Benamou-Brenier formula (Ambrosio et al., 2008, Equation 8.0.3) implies

$$W_2^2(\rho_0, \rho_1) = \inf_{(\rho_t, v_t): \, \partial_t \rho_t + \nabla \cdot (\rho_t v_t) = 0; \, \rho_{t=0} = \rho_0, \rho_{t=1} = \rho_1} \int_0^1 \|v_t\|_{L^2(\rho_t)}^2 \mathrm{d}t.$$

Thus, one can easily conclude the desired inequality.

To show that the inequality is attained when $(\rho_t)_{t \in [0,1]}$ is the constant-speed Wasserstein geodesic, note that the derivation implies that the inequality is attained when $|\dot{\rho}|_t$ is a constant for all $t \in [0, 1]$, and for any $0 \leq t_1 < t_2 < t_3 \leq 1$, $W_2(\rho_{t_1}, \rho_{t_2}) + W_2(\rho_{t_2}, \rho_{t_3}) = W_2(\rho_{t_1}, \rho_{t_3})$. We refer readers to Ambrosio et al. (2008, Chapter 7.2) for the construction of the constant-speed Wasserstein geodesic.

**2.** It is known from Bakry et al. (2014) that $\pi$ satisfies $C$-LSI implies

$$\mathrm{KL}(\mu \| \pi) \leq \frac{C}{2} \mathbb{E}_\mu \left\| \nabla \log \frac{\mathrm{d}\mu}{\mathrm{d}\pi} \right\|^2, \quad \mathrm{KL}(\mu \| \pi) \geq \frac{1}{2C} W_2^2(\mu, \pi), \; \forall \mu.$$

Therefore,

$$\frac{1}{\delta^2} W_2^2(\rho_{t+\delta}, \rho_t) \leq C_{\mathrm{LSI}}(\rho_t)^2 \int \frac{\|\nabla \log \rho_{t+\delta} - \nabla \log \rho_t\|^2}{\delta^2} \mathrm{d}\rho_{t+\delta}.$$

By letting $\delta \to 0$ and assuming regularity conditions, we have

$$|\dot{\rho}|_t^2 \leq C_{\mathrm{LSI}}(\rho_t)^2 \int \|\partial_t \nabla \log \rho_t\|^2 \mathrm{d}\rho_t = C_{\mathrm{LSI}}(\rho_t)^2 \|\partial_t \nabla \log \rho_t\|_{L^2(\rho_t)}^2$$

.

**3.** By Liu (2020), we know that $\pi$ satisfies $C$-PI implies

$$W_2^2(\mu, \pi) \le 2C \left( \int \frac{\mathrm{d}\mu}{\mathrm{d}\pi} \mathrm{d}\mu - 1 \right), \ \forall \mu.$$

Therefore,

$$\frac{1}{\delta^2} W_2^2(\rho_{t+\delta}, \rho_t) \le 2C_{\mathrm{PI}}(\rho_t) \int \frac{(\rho_{t+\delta} - \rho_t)^2}{\delta^2 \rho_t} \mathrm{d}x.$$

By letting $\delta \to 0$, we have

$$|\dot{\rho}|_t^2 \le 2C_{\mathrm{PI}}(\rho_t) \int \frac{(\partial_t \rho_t)^2}{\rho_t} \mathrm{d}x = 2C_{\mathrm{PI}}(\rho_t) \|\partial_t \log \rho_t\|_{L^2(\rho_t)}^2.$$

$\square$

**Proof of Ex. 1.**

**1.** The reparameterized curve $(p_s)_{s \in [0,S]}$ satisfies the standard heat equation:

$$\partial_s p_s = \Delta p_s \implies \partial_s p_s + \nabla \cdot (p_s(-\nabla \log p_s)) = 0,$$

which means $(-\nabla \log p_s)_{s \in [0,S]}$ generates $(p_s)_{s \in [0,S]}$. According to the uniqueness result in Lem. 2 and also Ambrosio et al. (2008, Theorem 8.3.1), the Lebesgue-a.e. unique vector field $(v_t^*)_{t \in [a,b]}$ that generates $(\rho_t)_{t \in [a,b]}$ and satisfies $|\dot{\rho}|_t = \|v_t^*\|_{L^2(\rho_t)}$ for Lebesgue-a.e. $t \in [a,b]$ can be written in a gradient field $v_t^* = \nabla \phi_t$ for some $\phi_t : \mathbb{R}^d \to \mathbb{R}$. This implies that

$$|\dot{p}|_s = \|\nabla \log p_s\|_{L^2(p_s)}^2.$$

By the time change $[0, S] \ni s \mapsto \theta = s/S \in [0, 1]$ and taking integral, we obtain the desired result.

**2.** Since $p_s = \sum_{i=1}^N w_i \mathcal{N}\left(\mu_i, (\sigma^2 + 2s)I\right)$, by Lem. 7, we know that $-\nabla^2 \log p_s \preceq \frac{1}{\sigma^2 + 2s}I$. Thus Lem. 6 implies $\|\nabla \log p_s\|_{L^2(p_s)}^2 \le \frac{d}{\sigma^2 + 2s}$, so

$$\mathcal{A} \le S \int_0^S \frac{d}{\sigma^2 + 2s} \mathrm{d}s = \frac{Sd}{2} \log\left(1 + \frac{2S}{\sigma^2}\right).$$

Finally, the LSI and PI constant of mixture of Gaussian distributions are, in general, exponential with respect to the maximum distance between the means. To illustrate this, consider a simpler case where there are only 2 Gaussian components: $\rho = \frac{1}{2}\mathcal{N}(0, \sigma^2 I) + \frac{1}{2}\mathcal{N}(y, \sigma^2 I)$. Schlichting (2019, Section 4.1) has shown that $C_{\mathrm{PI}}(\rho) \le C_{\mathrm{LSI}}(\rho) \le \frac{\sigma^2}{2}\left(\mathrm{e}^{\|y\|^2/\sigma^2} + 3\right)$, while Dong & Tong (2022, Proposition 1) has shown that when $\|y\| \gtrsim \sigma\sqrt{d}$, $C_{\mathrm{LSI}}(\rho) \ge C_{\mathrm{PI}}(\rho) \gtrsim \frac{\sigma^4}{\|y\|^2}\mathrm{e}^{\Omega(\|y\|^2/\sigma^2)}$. $\square$

## B  SAMPLING FROM $\pi_0$

**Lemma 5.** *When $\eta_0 = 0$, $\pi_0 = \mathcal{N}\left(0, \lambda_0^{-1}I\right)$ can be sampled directly; when $\eta_0 \in (0, 1]$, we choose $\lambda_0 = \eta_0 d\beta$, so that under Assump. 2, it takes $O\left(1 \vee \log \frac{\eta_0 \beta R^2}{d^2}\right) = \widetilde{O}(1)$ queries to the oracle of $V$ and $\nabla V$ in expectation to precisely sample from $\pi_0$ via rejection sampling.*

*Proof.* Our proof is inspired by Liang & Chen (2023). We only consider the nontrivial case $\eta_0 \in (0, 1]$. The potential of $\pi_0$ is $V_0 := \eta_0 V + \frac{\lambda_0}{2}\|\cdot\|^2$, which is $(\lambda_0 - \eta_0\beta)$-strongly-convex and $(\lambda_0 + \eta_0\beta)$-smooth. Note that for any fixed point $x' \in \mathbb{R}^d$,

$$\pi_0(x) \propto \exp\left(-V_0(x)\right) \le \exp\left(-V_0(x') - \langle \nabla V_0(x'), x - x' \rangle - \frac{\lambda_0 - \eta_0\beta}{2}\|x - x'\|^2\right).$$

The r.h.s. is the unnormalized density of $\pi_0' := \mathcal{N}\left(x' - \frac{\nabla V_0(x')}{\lambda_0 - \eta_0\beta}, \frac{1}{\lambda_0 - \eta_0\beta}I\right)$. The rejection sampling algorithm is as follows: sample $X \sim \pi_0'$, and accept $X$ as a sample from $\pi_0$ with probability

$$\exp\left(-V_0(X) + V_0(x') + \langle \nabla V_0(x'), X - x' \rangle + \frac{\lambda_0 - \eta_0\beta}{2}\|X - x'\|^2\right) \in (0, 1].$$

By Chewi (2024, Theorem 7.1.1), the number of queries to the oracle of $V$ until acceptance follows a geometric distribution with mean

$$\frac{\int \exp\left(-V_0(x') - \langle \nabla V_0(x'), x - x' \rangle - \frac{\lambda_0 - \eta_0\beta}{2}\|x - x'\|^2\right) \mathrm{d}x}{\int \exp\left(-V_0(x)\right) \mathrm{d}x}$$

$$\leq \frac{\int \exp\left(-V_0(x') - \langle \nabla V_0(x'), x - x' \rangle - \frac{\lambda_0 - \eta_0\beta}{2}\|x - x'\|^2\right) \mathrm{d}x}{\int \exp\left(-V_0(x') - \langle \nabla V_0(x'), x - x' \rangle - \frac{\lambda_0 + \eta_0\beta}{2}\|x - x'\|^2\right) \mathrm{d}x}$$

$$= \frac{\int \exp\left(-\langle \nabla V_0(x'), x \rangle - \frac{\lambda_0 - \eta_0\beta}{2}\|x\|^2\right) \mathrm{d}x}{\int \exp\left(-\langle \nabla V_0(x'), x \rangle - \frac{\lambda_0 + \eta_0\beta}{2}\|x\|^2\right) \mathrm{d}x}$$

$$= \frac{\int \exp\left(-\frac{\lambda_0 - \eta_0\beta}{2}\left\|x + \frac{\nabla V_0(x')}{\lambda_0 - \eta_0\beta}\right\|^2 + \frac{\|\nabla V_0(x')\|^2}{2(\lambda_0 - \eta_0\beta)}\right) \mathrm{d}x}{\int \exp\left(-\frac{\lambda_0 + \eta_0\beta}{2}\left\|x + \frac{\nabla V_0(x')}{\lambda_0 + \eta_0\beta}\right\|^2 + \frac{\|\nabla V_0(x')\|^2}{2(\lambda_0 + \eta_0\beta)}\right) \mathrm{d}x}$$

$$= \left(\frac{\lambda_0 + \eta_0\beta}{\lambda_0 - \eta_0\beta}\right)^{\frac{d}{2}} \exp\left(\frac{\eta_0\beta}{\lambda_0^2 - \eta_0^2\beta^2}\|\nabla V_0(x')\|^2\right)$$

$$\leq \exp\left(\frac{\eta_0\beta d}{\lambda_0 - \eta_0\beta}\right) \exp\left(\frac{\eta_0\beta}{\lambda_0^2 - \eta_0^2\beta^2}\|\nabla V_0(x')\|^2\right)$$

We choose $\lambda_0 = \eta_0\beta d$ such that $\exp\left(\frac{\eta_0\beta d}{\lambda_0 - \eta_0\beta}\right) \lesssim 1$. With this $\lambda_0$, $\exp\left(\frac{\eta_0\beta}{\lambda_0^2 - \eta_0^2\beta^2}\|\nabla V_0(x')\|^2\right)$ is also $O(1)$ as long as $\|\nabla V_0(x')\|^2 \lesssim \eta_0\beta d^2$.

Let $x''$ be the global minimizer of the strongly convex potential function $V_0$, which satisfies

$$0 = \nabla V_0(x'') = \eta_0 \nabla V(x'') + \lambda_0 x'' = \eta_0 \nabla V(x'') + \eta_0\beta d x''.$$

Given the smoothness of $V_0$,

$$\|\nabla V_0(x')\|^2 = \|\nabla V_0(x') - \nabla V_0(x'')\|^2 \leq (\eta_0\beta d + \eta_0\beta)^2 \|x' - x''\|^2.$$

Therefore, to guarantee $\|\nabla V_0(x')\|^2 \lesssim \eta_0\beta d^2$, it suffices to find an $x'$ such that $\|x' - x''\| \lesssim \frac{1}{\sqrt{\eta_0\beta}}$.

We first derive an upper bound of $\|x''\|$ under Assump. 2. Since $x_*$ is a global minimizer of $V$,

$$\beta d\|x''\| = \|\nabla V(x'')\| = \|\nabla V(x'') - \nabla V(x_*)\|$$
$$\leq \beta\|x'' - x_*\| \leq \beta(\|x''\| + R),$$
$$\implies \|x''\| \leq \frac{R}{d - 1} \lesssim \frac{R}{d}.$$

When $R = 0$, i.e., $0$ is a known global minimizer of both $V$ and $V_0$, we can directly set $x' = 0$; otherwise, we need to run optimization algorithms to find such an $x'$. According to standard results in convex optimization (see, e.g., Garrigos & Gower (2023, Theorem 3.6)), running gradient descent on function $V_0$ with step size $\frac{1}{\lambda_0 + \eta_0\beta}$ yields the following convergence rate: starting from $x_0 = 0$, the $t$-th iterate $x_t$ satisfies

$$\|x_t - x''\|^2 \leq \left(1 - \frac{\lambda_0 - \eta_0\beta}{\lambda_0 + \eta_0\beta}\right)^t \|0 - x''\|^2 \lesssim \left(\frac{2}{d}\right)^t \frac{R^2}{d^2} \leq \frac{R^2}{\mathrm{e}^t d^2},$$

where the last inequality holds when $d \geq 6$. Thus, $\log \frac{\eta_0\beta R^2}{d^2} + O(1)$ iterations are sufficient to find a desired $x'$. We summarize the rejection sampling in Alg. 2. In conclusion, precisely sampling from $\pi_0$ requires

$$O\left(1 \vee \log \frac{\eta_0\beta R^2}{d^2}\right) = \widetilde{O}(1)$$

calls to the oracle of $V$ and $\nabla V$ in expectation. $\qquad\square$

---

**Algorithm 2:** Rejection Sampling for $\pi_0$.

---

**Input:** $\pi \propto \exp(-V)$, $\eta_0 \in (0, 1]$, $\lambda_0 = \eta_0 \beta d$.

1 Let $V_0 := \eta_0 V + \frac{\lambda_0}{2} \|\cdot\|^2$;

2 Use gradient descent to find an $x'$ that is $O\left(\frac{1}{\sqrt{\eta_0 \beta}}\right)$-close to the global minimizer of $V_0$;

3 Let $\pi_0' := \mathcal{N}\left(x' - \frac{\nabla V_0(x')}{\lambda_0 - \eta_0 \beta}, \frac{1}{\lambda_0 - \eta_0 \beta} I\right)$;

4 **while** *True* **do**

5      Sample $X \sim \pi_0'$ and $U \in [0, 1]$ independently;

6      Accept $X$ as a sample of $\pi_0$ when

$$U \le \exp\left(-V_0(X) + V_0(x') + \langle \nabla V_0(x'), X - x' \rangle + \frac{\lambda_0 - \eta_0 \beta}{2} \|x - x'\|^2\right).$$

7 **end**

**Output:** $X \sim \pi_0$.

---

**Remark.** *The parameter $R$ reflects prior knowledge about global minimizer(s) of the potential function $V$. Unless it is exceptionally large, indicating scarce prior information about the global minimizer(s) of $V$, this $\widetilde{O}(1)$ complexity is negligible compared to the overall complexity of sampling. In particular, when the exact location of a global minimizer of $V$ is known, we can adjust the potential $V$ so that $0$ becomes a global minimizer, thereby reducing the complexity to $O(1)$.*

## C  PROOFS FOR ANNEALED LMC

### C.1  PROOF OF EQ. (7)

Define $\Lambda(t) := \int_0^t \lambda\left(\frac{\tau}{T}\right) d\tau$, whose derivative is $\Lambda'(t) = \lambda\left(\frac{t}{T}\right)$. By Itô's formula, we have

$$d\left(e^{\Lambda(t)} X_t\right) = e^{\Lambda(t)}\left(\lambda\left(\frac{t}{T}\right) X_t dt + dX_t\right) = e^{\Lambda(t)}\left(-\eta\left(\frac{t}{T}\right) \nabla V(X_{t_-}) dt + \sqrt{2} dB_t\right).$$

Integrating over $t \in [T_{\ell-1}, T_\ell)$ (note that in this case $t_- = T_{\ell-1}$), we obtain

$$e^{\Lambda(T_\ell)} X_{T_\ell} - e^{\Lambda(T_{\ell-1})} X_{T_{\ell-1}} = -\left(\int_{T_{\ell-1}}^{T_\ell} \eta\left(\frac{t}{T}\right) e^{\Lambda(t)} dt\right) \nabla V(X_{T_{\ell-1}}) + \sqrt{2} \int_{T_{\ell-1}}^{T_\ell} e^{\Lambda(t)} dB_t,$$

i.e.,

$$X_{T_\ell} = e^{-(\Lambda(T_\ell) - \Lambda(T_{\ell-1}))} X_{T_{\ell-1}} - \left(\int_{T_{\ell-1}}^{T_\ell} \eta\left(\frac{t}{T}\right) e^{-(\Lambda(T_\ell) - \Lambda(t))} dt\right) \nabla V(X_{T_{\ell-1}})$$

$$+ \sqrt{2} \int_{T_{\ell-1}}^{T_\ell} e^{-(\Lambda(T_\ell) - \Lambda(t))} dB_t.$$

Since

$$\Lambda(T_\ell) - \Lambda(t) = \int_t^{T_\ell} \lambda\left(\frac{\tau}{T}\right) d\tau = T \int_{t/T}^{T_\ell/T} \lambda(u) du,$$

by defining $\Lambda_0(\theta', \theta) = \exp\left(-T \int_\theta^{\theta'} \lambda(u) du\right)$, we have $e^{-(\Lambda(T_\ell) - \Lambda(T_{\ell-1}))} = \Lambda_0(\theta_\ell, \theta_{\ell-1})$. Similarly, we can show that

$$\int_{T_{\ell-1}}^{T_\ell} \eta\left(\frac{t}{T}\right) e^{-(\Lambda(T_\ell) - \Lambda(t))} dt = \int_{T_{\ell-1}}^{T_\ell} \eta\left(\frac{t}{T}\right) \Lambda_0\left(\frac{T_\ell}{T}, \frac{t}{T}\right) dt = T \int_{\theta_{\ell-1}}^{\theta_\ell} \eta(u) \Lambda_0(\theta_\ell, u) du,$$

and $\sqrt{2} \int_{T_{\ell-1}}^{T_\ell} e^{-(\Lambda(T_\ell) - \Lambda(t))} dB_t$ is a zero-mean Gaussian random vector with covariance

$$2 \int_{T_{\ell-1}}^{T_\ell} e^{-2(\Lambda(T_\ell) - \Lambda(t))} dt \cdot I = 2 \int_{T_{\ell-1}}^{T_\ell} \Lambda_0^2\left(\frac{T_\ell}{T}, \frac{t}{T}\right) dt \cdot I = 2T \int_{\theta_{\ell-1}}^{\theta_\ell} \Lambda_0^2(\theta_\ell, u) du \cdot I.$$

$\square$

## C.2 PROOF OF THM. 2

We denote the path measure of ALMC (Eq. (6)) by $\mathbb{Q}$. Then, $\mathbb{Q}_T$, the marginal distribution of $X_T$, serves as the output distribution $\nu^{\text{ALMC}}$. Similar to the methodology in the proof of Thm. 1, we use $\mathbb{P}$ to denote the reference path measure of Eq. (4), in which the vector field $(v_t)_{t\in[0,T]}$ generates the cure of probability distributions $(\widetilde{\pi}_t)_{t\in[0,T]}$.

Using the data-processing inequality, it suffices to demonstrate that $\mathrm{KL}(\mathbb{P}\|\mathbb{Q}) \leq \varepsilon^2$. By Girsanov theorem (Lem. 1) and triangle inequality, we have

$$
\begin{aligned}
\mathrm{KL}(\mathbb{P}\|\mathbb{Q}) &= \frac{1}{4}\int_0^T \mathbb{E}_{\mathbb{P}}\left\|\eta\left(\frac{t}{T}\right)\left(\nabla V(X_t) - \nabla V(X_{t_-})\right) - v_t(X_t)\right\|^2 \mathrm{d}t \\
&\lesssim \int_0^T \mathbb{E}_{\mathbb{P}}\left[\eta\left(\frac{t}{T}\right)^2 \|\nabla V(X_t) - \nabla V(X_{t_-})\|^2 + \|v_t(X_t)\|^2\right]\mathrm{d}t \\
&\leq \sum_{\ell=1}^M \eta\left(\frac{T_\ell}{T}\right)^2 \beta^2 \int_{T_{\ell-1}}^{T_\ell} \mathbb{E}_{\mathbb{P}}\|X_t - X_{t_-}\|^2 \mathrm{d}t + \int_0^T \|v_t\|_{L^2(\widetilde{\pi}_t)}^2 \mathrm{d}t.
\end{aligned}
$$

The last inequality arises from the smoothness of $V$ and the increasing property of $\eta(\cdot)$. To bound $\mathbb{E}_{\mathbb{P}}\|X_t - X_{t_-}\|^2$, note that under $\mathbb{P}$, for $t \in [T_{\ell-1}, T_\ell)$, we have

$$
X_t - X_{t_-} = \int_{T_{\ell-1}}^t \left(\nabla\log\widetilde{\pi}_\tau + v_\tau\right)(X_\tau)\mathrm{d}\tau + \sqrt{2}(B_t - B_{T_{\ell-1}}).
$$

Thanks to the fact that $X_t \sim \widetilde{\pi}_t$ under $\mathbb{P}$,

$$
\begin{aligned}
\mathbb{E}_{\mathbb{P}}\|X_t - X_{t_-}\|^2 &\lesssim \mathbb{E}_{\mathbb{P}}\left\|\int_{T_{\ell-1}}^t \left(\nabla\log\widetilde{\pi}_\tau + v_\tau\right)(X_\tau)\mathrm{d}\tau\right\|^2 + \mathbb{E}\|\sqrt{2}(B_t - B_{T_{\ell-1}})\|^2 \\
&\lesssim (t - T_{\ell-1})\int_{T_{\ell-1}}^t \mathbb{E}_{\mathbb{P}}\|(\nabla\log\widetilde{\pi}_\tau + v_\tau)(X_\tau)\|^2\mathrm{d}\tau + d(t - T_{\ell-1}) \\
&\lesssim (t - T_{\ell-1})\int_{T_{\ell-1}}^t \left(\|\nabla\log\widetilde{\pi}_\tau\|_{L^2(\widetilde{\pi}_\tau)}^2 + \|v_\tau\|_{L^2(\widetilde{\pi}_\tau)}^2\right)\mathrm{d}\tau + d(t - T_{\ell-1}) \\
&\lesssim h_\ell \int_{T_{\ell-1}}^{T_\ell} \left(\|\nabla\log\widetilde{\pi}_\tau\|_{L^2(\widetilde{\pi}_\tau)}^2 + \|v_\tau\|_{L^2(\widetilde{\pi}_\tau)}^2\right)\mathrm{d}\tau + dh_\ell.
\end{aligned}
$$

The second inequality arises from the application of the Cauchy-Schwarz inequality, and the last inequality is due to the definition $h_\ell = T_\ell - T_{\ell-1}$. Taking integral over $t \in [T_{\ell-1}, T_\ell]$,

$$
\int_{T_{\ell-1}}^{T_\ell} \mathbb{E}\|X_t - X_{t_-}\|^2 \mathrm{d}t \lesssim h_\ell^2 \int_{T_{\ell-1}}^{T_\ell} \left(\|\nabla\log\widetilde{\pi}_t\|_{L^2(\widetilde{\pi}_t)}^2 + \|v_t\|_{L^2(\widetilde{\pi}_t)}^2\right)\mathrm{d}t + dh_\ell^2.
$$

Recall that the potential of $\widetilde{\pi}_t$ is $\left(\eta\left(\frac{t}{T}\right)\beta + \lambda\left(\frac{t}{T}\right)\right)$-smooth. By Lem. 6 and the monotonicity of $\eta(\cdot)$ and $\lambda(\cdot)$, we have

$$
\begin{aligned}
\int_{T_{\ell-1}}^{T_\ell} \|\nabla\log\widetilde{\pi}_t\|_{L^2(\widetilde{\pi}_t)}^2 \mathrm{d}t &\leq \int_{T_{\ell-1}}^{T_\ell} d\left(\eta\left(\frac{t}{T}\right)\beta + \lambda\left(\frac{t}{T}\right)\right)\mathrm{d}t \\
&\leq dh_\ell\left(\beta\eta\left(\frac{T_\ell}{T}\right) + \lambda\left(\frac{T_{\ell-1}}{T}\right)\right) \\
&= dh_\ell\left(\beta\eta\left(\theta_\ell\right) + \lambda\left(\theta_{\ell-1}\right)\right).
\end{aligned}
$$

Therefore, $\mathrm{KL}(\mathbb{P}\|\mathbb{Q})$ is upper bounded by

$$
\lesssim \sum_{\ell=1}^{M} \eta\left(\theta_\ell\right)^2 \beta^2 \int_{T_{\ell-1}}^{T_\ell} \mathbb{E}_{\mathbb{P}}\|X_t - X_{t_-}\|^2 \mathrm{d}t + \int_0^T \|v_t\|_{L^2(\widetilde{\pi}_t)}^2 \mathrm{d}t
$$

$$
\lesssim \sum_{\ell=1}^{M} \eta\left(\theta_\ell\right)^2 \beta^2 \left( dh_\ell^3 \left(\beta\eta\left(\theta_\ell\right) + \lambda\left(\theta_{\ell-1}\right)\right) + h_\ell^2 \int_{T_{\ell-1}}^{T_\ell} \|v_t\|_{L^2(\widetilde{\pi}_t)}^2 \mathrm{d}t + dh_\ell^2 \right) + \int_0^T \|v_t\|_{L^2(\widetilde{\pi}_t)}^2 \mathrm{d}t
$$

$$
= \sum_{\ell=1}^{M} \left( \left(1 + \eta(\theta_\ell)^2 \beta^2 h_\ell^2\right) \int_{T_{\ell-1}}^{T_\ell} \|v_t\|_{L^2(\widetilde{\pi}_t)}^2 \mathrm{d}t + \eta(\theta_\ell)^2 \beta^2 dh_\ell^2 \left(1 + h_\ell \left(\beta\eta(\theta_\ell) + \lambda(\theta_{\ell-1})\right)\right) \right).
$$

For the remaining integral, given that $(v_t)_{t \in [0,T]}$ generates $(\widetilde{\pi}_t)_{t \in [0,T]}$, according to Lem. 2, we may choose $v_t$ such that $\|v_t\|_{L^2(\widetilde{\pi}_t)} = |\dot{\widetilde{\pi}}|_t$. Thus,

$$
\int_{T_{\ell-1}}^{T_\ell} \|v_t\|_{L^2(\widetilde{\pi}_t)}^2 \mathrm{d}t = \int_{T_{\ell-1}}^{T_\ell} |\dot{\widetilde{\pi}}|_t^2 \mathrm{d}t = \int_{T_{\ell-1}}^{T_\ell} \frac{1}{T^2} |\dot{\pi}|_{t/T}^2 \mathrm{d}t = \frac{1}{T} \int_{\theta_{\ell-1}}^{\theta_\ell} |\dot{\pi}|_\theta^2 \mathrm{d}\theta,
$$

through a change-of-variable analogous to that used in the proof of Thm. 1. Therefore,

$$
\mathrm{KL}(\mathbb{P}\|\mathbb{Q}) \lesssim \sum_{\ell=1}^{M} \left( \frac{1 + \eta(\theta_\ell)^2 \beta^2 h_\ell^2}{T} \int_{\theta_{\ell-1}}^{\theta_\ell} |\dot{\pi}|_\theta^2 \mathrm{d}\theta + \eta(\theta_\ell)^2 \beta^2 dh_\ell^2 \left(1 + h_\ell \left(\beta\eta(\theta_\ell) + \lambda(\theta_{\ell-1})\right)\right) \right).
$$

Assume $h_\ell \lesssim \frac{1}{\beta d}$ (which will be verified later), so we can further simplify the above expression to

$$
\mathrm{KL}(\mathbb{P}\|\mathbb{Q}) \lesssim \sum_{\ell=1}^{M} \left( \frac{1}{T} \int_{\theta_{\ell-1}}^{\theta_\ell} |\dot{\pi}|_\theta^2 \mathrm{d}\theta + \eta(\theta_\ell)^2 \beta^2 dh_\ell^2 \right)
$$

$$
= \frac{\mathcal{A}}{T} + \beta^2 d \sum_{\ell=1}^{M} \eta(\theta_\ell)^2 h_\ell^2
$$

$$
= \frac{\mathcal{A}}{T} + \beta^2 d \sum_{\ell=1}^{M} \eta(\theta_\ell)^2 T^2 (\theta_\ell - \theta_{\ell-1})^2.
$$

To bound the above expression by $\varepsilon^2$, we first select $T \asymp \frac{\mathcal{A}}{\varepsilon^2}$, mirroring the total time $T$ required for running annealed LD as specified in Thm. 1. This guarantees that the continuous dynamics closely approximate the reference path measure. Given that $\eta(\cdot) \le 1$, it remains only to ensure

$$
\beta^2 d \sum_{\ell=1}^{M} \eta(\theta_\ell)^2 T^2 (\theta_\ell - \theta_{\ell-1})^2 \le \beta^2 d \frac{\mathcal{A}^2}{\varepsilon^4} \sum_{\ell=1}^{M} (\theta_\ell - \theta_{\ell-1})^2 \lesssim \varepsilon^2,
$$

which is equivalent to

$$
\sum_{\ell=1}^{M} (\theta_\ell - \theta_{\ell-1})^2 \lesssim \frac{\varepsilon^6}{d\beta^2 \mathcal{A}^2}. \tag{8}
$$

To minimize $M$, we apply Cauchy-Schwarz inequality:

$$
\left( \sum_{\ell=1}^{M} 1 \right) \left( \sum_{\ell=1}^{M} (\theta_\ell - \theta_{\ell-1})^2 \right) \ge \left( \sum_{\ell=1}^{M} (\theta_\ell - \theta_{\ell-1}) \right)^2 = 1,
$$

which establishes a lower bound for $M$. The equality is achieved when $\theta_\ell - \theta_{\ell-1} = \frac{1}{M}$ for all $\ell \in [\![1, M]\!]$. Thus, selecting

$$
M \asymp \frac{d\beta^2 \mathcal{A}^2}{\varepsilon^6}
$$

satisfies the constraint given in Eq. (8). In this case, the step size $h_\ell = \frac{T}{M} \asymp \frac{\varepsilon^4}{d\beta^2} \lesssim \frac{1}{\beta d}$. Combining this with the $\widetilde{O}(1)$ complexity for sampling from $\pi_0$, we have completed the proof. $\square$

## D  PROOF OF EX. 2

The smoothness of $V$ comes from Lem. 7.

Note that $\pi(x) \propto \sum_{i=1}^{N} p_i \exp\left(-\frac{\beta}{2}\|x - y_i\|^2\right)$, and define

$$\widehat{\pi}_\lambda(x) \propto \pi(x) \exp\left(-\frac{\lambda}{2}\|x\|^2\right)$$

$$= \sum_{i=1}^{N} p_i \exp\left(-\frac{\lambda\beta}{2(\lambda+\beta)}\|y_i\|^2 - \frac{\lambda+\beta}{2}\left\|x - \frac{\beta}{\lambda+\beta}y_i\right\|^2\right)$$

$$\propto \sum_{i=1}^{N} p_i \exp\left(-\frac{\lambda+\beta}{2}\left\|x - \frac{\beta}{\lambda+\beta}y_i\right\|^2\right)$$

$$= \sum_{i=1}^{N} p_i \mathcal{N}\left(\frac{\beta}{\lambda+\beta}y_i, \frac{1}{\lambda+\beta}I\right).$$

We use coupling method to upper bound $W_2^2(\widehat{\pi}_\lambda, \widehat{\pi}_{\lambda+\delta})$. We first sample $I$ following distribution $\mathbb{P}(I = i) = p_i, i \in [\![1, N]\!]$, and then independently sample $\eta \sim \mathcal{N}(0, I)$. We have

$$X := \frac{\beta}{\lambda+\beta}y_I + \frac{1}{\sqrt{\lambda+\beta}}\eta \sim \widehat{\pi}_\lambda,$$

$$Y := \frac{\beta}{\lambda+\delta+\beta}y_I + \frac{1}{\sqrt{\lambda+\delta+\beta}}\eta \sim \widehat{\pi}_{\lambda+\delta}.$$

By the definition of the $W_2$ distance, we have

$$W_2^2(\widehat{\pi}_\lambda, \widehat{\pi}_{\lambda+\delta}) \le \mathbb{E}\|X - Y\|^2$$

$$= \mathbb{E}_I \mathbb{E}_\eta \left\|\left(\frac{\beta}{\lambda+\beta} - \frac{\beta}{\lambda+\delta+\beta}\right)y_I + \left(\frac{1}{\sqrt{\lambda+\beta}} - \frac{1}{\sqrt{\lambda+\delta+\beta}}\right)\eta\right\|^2$$

$$= \mathbb{E}_I \left(\frac{\beta}{\lambda+\beta} - \frac{\beta}{\lambda+\delta+\beta}\right)^2\|y_I\|^2 + \left(\frac{1}{\sqrt{\lambda+\beta}} - \frac{1}{\sqrt{\lambda+\delta+\beta}}\right)^2 d$$

$$= \left(\frac{\beta}{\lambda+\beta} - \frac{\beta}{\lambda+\delta+\beta}\right)^2 r^2 + \left(\frac{1}{\sqrt{\lambda+\beta}} - \frac{1}{\sqrt{\lambda+\delta+\beta}}\right)^2 d.$$

This implies

$$|\dot{\widehat{\pi}}|_\lambda^2 = \lim_{\delta \to 0} \frac{W_2^2(\widehat{\pi}_\lambda, \widehat{\pi}_{\lambda+\delta})}{\delta^2} \le \frac{\beta^2 r^2}{(\lambda+\beta)^4} + \frac{d}{4(\lambda+\beta)^3}.$$

By time reparameterization $\pi_\theta = \widehat{\pi}_{\lambda(\theta)}$, $|\dot{\pi}|_\theta = |\dot{\widehat{\pi}}|_{\lambda(\theta)}|\dot{\lambda}(\theta)|$. With $\lambda(\theta) = \lambda_0(1-\theta)^\gamma$, $|\dot{\lambda}(\theta)| = \gamma\lambda_0(1-\theta)^{\gamma-1} \le \gamma\lambda_0 \lesssim \lambda_0$. Therefore,

$$\mathcal{A} = \int_0^1 |\dot{\pi}|_\theta^2 d\theta = \int_0^1 |\dot{\widehat{\pi}}|_{\lambda(\theta)}^2 |\dot{\lambda}(\theta)|^2 d\theta$$

$$\lesssim \lambda_0 \int_0^1 |\dot{\widehat{\pi}}|_{\lambda(\theta)}^2 |\dot{\lambda}(\theta)| d\theta = \lambda_0 \int_0^{\lambda_0} |\dot{\widehat{\pi}}|_\lambda^2 d\lambda$$

$$= \lambda_0 \int_0^{\lambda_0} \left(\frac{\beta^2 r^2}{(\lambda+\beta)^4} + \frac{d}{4(\lambda+\beta)^3}\right) d\lambda$$

$$\lesssim \lambda_0 \left(\beta^2 r^2 \frac{1}{\beta^3} + d\frac{1}{\beta^2}\right)$$

$$\lesssim d\beta(r^2\beta + 1)\left(\frac{r^2}{\beta} + \frac{d}{\beta^2}\right)$$

$$= d(r^2\beta + 1)\left(r^2 + \frac{d}{\beta}\right).$$

It follows from the proof that as long as $\max_{\theta \in [0,1]} |\dot{\lambda}(\theta)|$ is bounded by a polynomial function of $\beta$, $d$ and $r$, so is the action $\mathcal{A}$. $\qquad\square$

## E  SUPPLEMENTARY LEMMAS

**Lemma 6** (Chewi (2024, Lemma 4.E.1)). *Consider a probability measure $\mu \propto e^{-U}$ on $\mathbb{R}^d$. If $\nabla^2 U \preceq \beta I$ for some $\beta > 0$, then $\mathbb{E}_\mu \|\nabla U\|^2 \leq \beta d$.*

**Lemma 7** (Cheng et al. (2023, Lemma 4)). *For a Gaussian mixture distribution $\pi = \sum_i w_i \mathcal{N}\left(\mu_i, \sigma^2 I\right)$, we have*

$$\left(\frac{1}{\sigma^2} - \frac{\max_{i,j} \|\mu_i - \mu_j\|^2}{2\sigma^4}\right) I \preceq -\nabla^2 \log \pi \preceq \frac{1}{\sigma^2} I.$$

**Remark.** *This is a slightly improved version, as in the original lemma the authors omitted the factor 2 in the denominator on the l.h.s.*

**Lemma 8.** *Under Assump. 2, $\pi_\theta$ defined in Eq. (5) has finite second-order moment when $\eta(\theta) \in [0, 1]$.*

*Proof.* When $\eta(\theta) = 1$, $\pi_\theta \propto \exp\left(-V - \frac{\lambda(\theta)}{2}\|\cdot\|^2\right) \leq \exp(-V)$, and the claim is straightforward. Otherwise, by convexity of $u \mapsto e^{-u}$, we have

$$\exp\left(-\eta(\theta)V - \frac{\lambda(\theta)}{2}\|\cdot\|^2\right) = \exp\left(-\eta(\theta)V - (1 - \eta(\theta))\frac{\lambda(\theta)}{2(1 - \eta(\theta))}\|\cdot\|^2\right)$$

$$\leq \eta(\theta)\exp(-V) + (1 - \eta(\theta))\exp\left(-\frac{\lambda(\theta)}{2(1 - \eta(\theta))}\|\cdot\|^2\right).$$

Multiplying both sides by $\|\cdot\|^2$ and taking integral over $\mathbb{R}^d$, we see that $\mathbb{E}_{\pi_\theta}\|\cdot\|^2 < +\infty$.  □

## F  FURTHER DETAILS OF EXPERIMENTS IN SEC. 6

**Hyperparameter settings.** For all the experiments, we use the annealing schedule $\lambda(\theta) = 5(1 - \theta)^{10}$ and $\eta(\theta) \equiv 1$. The step size for ALMC is designed to follow a quadratic schedule: the step size at the $\ell$-th iteration (out of $M$ total iterations, $\ell \in [\![1, M]\!]$) is given by $-\frac{s_{\max} - s_{\min}}{M^2/4}\left(\ell - \frac{M}{2}\right)^2 + s_{\max}$, which increases on $\left[0, \frac{M}{2}\right]$ and decreases on $\left[\frac{M}{2}, M\right]$, with a maximum of $s_{\max}$ and a minimum of $s_{\min}$. For $r = 2, 5, 10, 15, 20, 25, 30$, the $M$ we choose are $200, 500, 2500, 10000, 20000, 40000, 60000$, respectively, and we set $s_{\max} = 0.05$ and $s_{\min} = 0.01$, which achieve the best performance across all settings after tuning.

**KL divergence estimation.** We use the Information Theoretical Estimators (ITE) toolbox (Szabó, 2014) to empirically estimate the KL divergence. In all cases, we generate 1000 samples using ALMC, and an additional 1000 samples from the target distribution. The KL divergence is estimated using the `ite.cost.BDKL_KnnK()` function, which leverages $k$-nearest-neighbor techniques to compute the divergence from the target distribution to the sampled distribution.

**Regression coefficients.** We also compute the linear regression approximation of the curves in Fig. 2 via `sklearn.linear_model.LinearRegression`. The blue curve has a slope of 2.841 and an intercept of 1.257, while the orange one has a slope of 2.890 and an intercept of 0.904. The $R^2$ scores, calculated using `sklearn.metrics.r2_score`, are 0.995 and 0.997, respectively.

