# OpenReview forum: "Provable Benefit of Annealed Langevin Monte Carlo for Non-log-concave Sampling"
_ICLR.cc/2025/Conference — ICLR 2025 Poster_

### Official Review · Reviewer_WbhJ · 2024-10-29

**Soundness:** 3
**Presentation:** 3
**Contribution:** 3
**Rating:** 8
**Confidence:** 4

**Summary:**

The paper consider the problem of sampling from an unnormalized density that may be non-log-concave and multimodal using annealed MCMC.  The authors present a non-asymptotic analysis of Annealed Langevin Monte Carlo, bypassing the commonly assumed log-concavity constraint in literature. The quantitative bounds derived are widely applicable and either comparable to or better than previous analyses for isoperimetry-free sampling methods.

**Strengths:**

The paper provides, as far as I know, the first thorough and non-asymptotic analysis of the annealed Langevin Monte Carlo algorithm.  The results obtained advance the understanding and analysis of annealed MCMC, potentially setting the stage for further research in this area. The intuition behind the algorithm and the proofs are well explained.

**Weaknesses:**

The discussion on the impact of the annealing schedule on the convergence rate is not sufficiently clear in the paper. How the choice of the annealing schedule influences the derived bounds? Specifically, in the example of mixture of Gaussians,  $\lambda(\theta) \propto (1-\theta)^\gamma$ is used,   how varying $\gamma$ affects the action $\mathcal{A}$? whether there is an optimal value of $\gamma$ that could minimize the convergence rate?

**Questions:**

Please see the weakness part.

---

> ### Author Response · Authors · 2024-11-21
> **Response to reviewer WbhJ**
>
> Thank you for your appreciation for our work.
>
> **Weakness**: *How the choice of the annealing schedule influences the derived bounds? Specifically, in the example of mixture of Gaussians, $\lambda(\theta) \propto (1-\theta)^\gamma$ is used, how varying $\gamma$ affects the action $\cal A$? whether there is an optimal value of $\gamma$ that could minimize the convergence rate?*
>
> **Response**: In this example, we obtain a uniform upper bound of the action $\cal A$ for all $\gamma\in[1,O(1)]$. We choose $\gamma\ge 1$ as intuitively, when $\theta$ is close to $1$, the distribution $\pi_\theta$ is more diffucult to sample from, and we should move more slowly. We agree that for each problem, there should be an optimal value of the parameter $\gamma$ that reaches the best sampling quality, yet studying the influence of $\gamma$ on the action $\cal A$ is highly non-trivial. This will be the focus of our study in the future.
>
> We hope that the response above could successfully address your concerns.

---

### Official Review · Reviewer_WqFj · 2024-10-31

**Soundness:** 2
**Presentation:** 3
**Contribution:** 3
**Rating:** 6
**Confidence:** 4

**Summary:**

The paper proposes a novel analysis of Annealed LMC which uses a specific scheduling system. The latter allows to benefit from another analysis scheme which uses the action on curves instead of isoperimetric inequalities.

**Strengths:**

The paper provides a concise analysis with a clever mathematical trick of using the action of the curve instead of the isoperimetry constants. Assuming gradient Lipschitzness of the potential function allows to easily (rejection-acceptance) sample from an initial distribution, which then allows to sample from the target distribution (non-convex, gradient-Lipschitz) using an annealing schedule for LMC.

**Weaknesses:**

- Assumption 1 is not well discussed. One cannot verify beforehand whether the curve $\pi_{\theta}$ is AC. How should it be done? See the respective question in the section below.
- Assumption 2 is not the classic smoothness assumption. Instead it is the gradient Lipschitz continuity, which is stronger. This should be explicitly mentioned in the paper.
- The notation is slightly confusing. There are too many indices for $\pi$ and the usage is incoherent. The reading would be rather simplified, if the notation was more developed.
- Simple experiments on sampling from a mixture of Gaussians using the proposed and existing methods would significantly boost the paper.
- In the abstract of the paper, the authors claim that their analysis is non-asymptotic, however it is not true. Theorem 2 (main result) and related results are given with $\tilde{O}(\cdot)$.
### Mathematical comments

- *The equation after line 1077* in the proof of Example 2 is incorrect. The density of the sum of two independent random variables(vectors) is the convolution of their densities rather than their product. Thus, the variable $X$ is not distributed as $\hat{\pi}_{\lambda}$. This serves as the cornerstone of the remainder of the proof, which means that the proof is incorrect.
- *The third line in the proof of Example 2* is incorrect. After recentering the Gaussians, the authors erase the terms $\exp(-\frac{2\lambda \beta}{\lambda + \beta})\|y_i\|^2$ with the $\propto$ sign. Since we have different $y_i$'s, the resulting mixture weights will be different from $p_i$. Nevertheless, this could be fixed by replacing $p_i$ by some $q_i$'s.
- *Paragraph starting at line 364* The authors motivate the use of annealing as the schedules start at a distribution $\pi_0$, which is easy to sample from. The reason for the easy sampling according to the authors is the strong-convexity of the negative log-density $-\log\pi_0(x) = \eta(0)V(x) + \lambda_0\|x\|^2/2$. The latter is true when $\lambda_0$ is large enough.
- *Continuation of the previous point. Line 831* The strong convexity constant $(\lambda_0 - \eta_0 \beta)$ might be negative, unless $\lambda_0$ is chosen to be large.
- *Beginning of the proof of Theorem 1* $v_t$ is chosen to satisfy the Fokker-Planck equation on line 294. However, later on line 301 it is chosen to minimize the norm $\|v_t\|_{L_2(\tilde{\pi}_t)}$  using Lemma 2. This is possible if $v_t$ also generates $\tilde{\pi}_t$. Is it possible to find such $v_t$? See the relevant question.

#### typos
- *line 956:* cure -> curve

**Questions:**

- Is the analysis using the action only available for annealing? Can we gain anything by applying this technique to LMC in the standard setting?
- What are the estimates of the action parameter $\mathcal{A}$ for a more general class of distributions than the mixture of Gaussians?
- Is it possible to guarantee that for certain target distributions and schedules Assumption 1 is satisfied? This is important, as it is not easy to manually check for every setting.
- Is there always a $v_t$ such that $X_t \sim \tilde{\pi}_t$ for all $t$ in (4) and it generates $\tilde{\pi}_t$ i.e. satisfies the Fokker-Planck equation?

---

> ### Author Response · Authors · 2024-11-21
> **Response to reviewer WqFj (1)**
>
> Thank you for your careful reading of our paper and your valuable, constructive feedback.
>
> **Weakness 1 and Question 3**: *One cannot verify beforehand whether the curve is AC. Is it possible to guarantee that for certain target distributions and schedules Assumption 1 is satisfied?*
>
> **Response**: Absolute continuity is a fairly weak regularity condition on the curve of probability measures. Loosely speaking, as long as the curve $(\pi _\theta) _{\theta\in[0,1]}$ has a density $\pi _\theta(x)>0$ that is jointly $C^1$ w.r.t both $x\in\mathbb{R}^d$ and $\theta\in[0,1]$, then it is AC.
>
> For instance, we can intuitively prove the proposition above in one-dimension. Let $F _\theta(x)=\int _{-\infty}^x\pi _\theta(u){\rm d}u$ be the c.d.f. of $\pi _\theta$, which is strictly increasing, and its inverse $F _\theta^{-1}$ is well-defined. $F^{-1} _{\cdot}(\cdot)$ is also jointly $C^1$. We know from standard textbook of OT that
> $$W _2^2(\pi _\theta,\pi _{\theta+\delta})=\int _0^1|F _{\theta+\delta}^{-1}(q)-F _\theta^{-1}(q)|^2{\rm d}q.$$
> The above quantity should be $O(\delta^2)$ as $\delta\to0$ due to the $C^1$ property, and hence the curve is AC.
>
> **Weakness 2**: *Assumption 2 is not the classic smoothness assumption. Instead it is the gradient Lipschitz continuity, which is stronger. This should be explicitly mentioned in the paper.*
>
> **Response**: We agree that smoothness has several different definitions, such as being $C^1$ or $C^\infty$. The definition we used in this paper, the Lipschitz continuity of the gradient (see the "notations and definitions" part in Sec. 1), is also a common one that appears in many standard textbooks in optimization and sampling, such as Sec. 3.2 of Bubeck's [Convex Optimization: Algorithms and Complexity](https://dl.acm.org/doi/10.1561/2200000050) and Chap. 4 of Chewi's [Log-concave Sampling](https://chewisinho.github.io/main.pdf). We appreciate your comment for the potential ambiguity.
>
> **Weakness 3**: *There are too many indices for $\pi$ and the usage is incoherent.*
>
> **Response**: Sorry about the confusion the paper might have caused you. Throughout this paper, $(\pi _\theta) _{\theta\in[0,1]}$ is an AC curve bridging $\pi _0$, a simple distribution, and $\pi _1$, the target distribution, while $(\widetilde{\pi} _t=\pi _{t/T}) _{t\in[0,T]}$ is the curve after time-reparameterization. $0=\theta _0<\theta _1<...<\theta _M=1$ are the discrete time points on $[0,1]$, and $T _\ell=T\theta _\ell$'s are the discrete time points on $[0,T]$, with $h _\ell=T(\theta _\ell-\theta _{\ell-1})$ being the step size. We hope that this clarifies the usage of these notations.
>
> **Weakness 4**: *Simple experiments on sampling from a mixture of Gaussians using the proposed and existing methods would significantly boost the paper.*
>
> **Response**: Please see Sec. 6 for the experimental results, where we verified the findings in Ex. 2.

---

> > ### Author Response · Authors · 2024-11-21
> > **Response to reviewer WqFj (2)**
> >
> > **Weakness 5**: *In the abstract of the paper, the authors claim that their analysis is non-asymptotic, however it is not true. Theorem 2 (main result) and related results are given with $\widetilde{O}(\cdot)$.*
> >
> > **Response**: "Non-asymptotic analysis" refers to providing explicit bounds on algorithmic performance for fixed problem parameters, such as dimension $d$, accuracy $\varepsilon$, and smoothness parameter $\beta$, rather than focusing on behavior in asymptotic regimes.
> >
> > To better explain the difference between asymptotic and non-asymptotic results, consider the following two examples:
> >
> > 1. Central limit theorem (CLT) implies that for i.i.d. copies $X _1,X _2,...$ of a random variable $X$ with mean $\mu$ and variance $\sigma^2$, the law of $\sqrt{n}(\overline X _n-\mu)$ converges to ${\cal N}(0,\sigma^2)$ as $n\to\infty$, where $\overline X _n$ is the mean of $X _1,...,X _n$. Here, we only know the limiting behavior of the law of $\sqrt{n}(\overline X _n-\mu)$ as $n\to\infty$; for any finite $n$, CLT does not imply the discrepancy (say, KL divergence, TV distance, etc.) between the law of $\sqrt{n}(\overline X _n-\mu)$ and ${\cal N}(0,\sigma^2)$. Hence, this result is an example of **asymptotic analysis**.
> > 2. In optimization, we know that gradient descent has linear convergence for strongly convex and smooth function. More precisely, suppose $f$ is $\alpha$-strongly-convex and $\beta$-smooth with global minimizer $x _\star$, then with step size $\frac{1}{\beta}$, $\|x _n-x _\star\|^2\le\left(1-\frac{\alpha}{\beta}\right)^n\|x _0-x _\star\|^2$. This result not only implies that as the number of iterate $n\to\infty$, the iterate $x _n$ converges to $x _\star$, but also tells us how far we are at every finite step $n$, and how many iterations are required to reach a point that is $\varepsilon$-close to the global minimizer, which is upper bounded by $\frac{\beta}{\alpha}\log\frac{\|x _0-x _\star\|^2}{\varepsilon^2}$, i.e., $O\left(\frac{\beta}{\alpha}\log\frac{\|x _0-x _\star\|}{\varepsilon}\right)$. Hence, this result is an example of **non-asymptotic analysis**.
> >
> > In Thm. 2, we establish the number of oracle calls required to obtain samples that are $\varepsilon$-accuracy in KL divergence, which is clearly an example of **non-asymptotic analysis**. The asymptotic notations $O(\cdot)$ and $\widetilde O(\cdot)$ are only used to hide constant and logarithmic terms. We hope this clarifies the difference between asymptotic and non-asymptotic analysis.
> >
> > **Mathematical comment 1**: *The equation after line 1077 in the proof of Example 2 is incorrect.*
> >
> > **Response**: Thank you for your careful reading of our proof and for pointing out this potential issue. However, upon careful review, we believe that the proof as presented in the paper is correct. Here, $\widehat{\pi} _\lambda=\sum _{i=1}^N p _i{\cal N}\left(\frac{\beta}{\lambda+\beta}y _i,\frac{1}{\lambda+\beta}I\right)$. To sample from such a mixture of Gaussians with the same covariance, we can first sample the means $\frac{\beta}{\lambda+\beta}y _i$ w.p. $p _i$, then convolute it with ${\cal N}\left(0,\frac{1}{\lambda+\beta}I\right)$. Our constructions of $X$ is formed by picking the mean and convoluting it with Gaussian, which should be correct.
> >
> > **Mathematical comment 2**: *The third line in the proof of Example 2 is incorrect.*
> >
> > **Response**: Please note that we have assumed all $y _i$'s have the same norm $r$ in this example, and hence all $\exp\left(-\frac{2\lambda \beta}{\lambda + \beta}\right)\|y _i\|^2$'s are the same and can be cancelled.
> >
> > **Mathematical comments 3 and 4**: *Paragraph starting at line 364. [...] The latter is true when $\lambda$ is large enough. Line 831 The strong convexity constant $(\lambda _0 - \eta _0 \beta)$ might be negative, unless $\lambda _0$ is chosen to be large.*
> >
> > **Response**: Your observation is correct. In Lem. 4, we have proved that with $\lambda _0 = \eta _0d\beta$, the complexity of sampling from $\pi _0$ is $\widetilde{O}(1)$.

---

> > > ### Author Response · Authors · 2024-11-21
> > > **Response to reviewer WqFj (3)**
> > >
> > > **Mathematical comment 5 & Question 4**: *Beginning of the proof of Theorem 1. [...] Is there always a $v _t$ such that $X _t \sim \tilde{\pi} _t$ for all $t$ in (4) and it generates $\tilde{\pi} _t$ i.e. satisfies the Fokker-Planck equation?*
> > >
> > > **Response**: The logic here is as follows. We first choose any vector field $(v _t)$ s.t. it generates $(\widetilde\pi _t)$. Such vector field exists given the absolute continuity of $(\widetilde\pi _t)$, see, e.g., Thm. 8.3.1 of [Ambrosio et al., 2008](https://link.springer.com/book/10.1007/978-3-7643-8722-8) or Thm. 5.14 of [Santambrogio, 2015](https://link.springer.com/book/10.1007/978-3-319-20828-2). By the Fokker-Planck equation, this guarantees that $X _t\sim\widetilde{\pi} _t$ for all $t$ in (4). Girsanov theorem implies that ${\rm KL}(\mathbb{P}\|\mathbb{Q})=\frac{1}{4}\int _0^T\|v _t\| _{L^2(\widetilde\pi _t)}^2{\rm d}t$. Using Lem. 2, among all vector field $(v _t)$ that generates $(\widetilde{\pi} _t)$, we may choose the one that minimizes $\|v _t\| _{L^2(\widetilde\pi _t)}^2$, which implies that ${\rm KL}(\mathbb{P}\|\mathbb{Q})=\frac{\cal A}{4T}$.
> > >
> > > **Questions 1**: *Is the analysis using the action only available for annealing? Can we gain anything by applying this technique to LMC in the standard setting?*
> > >
> > > **Response**: We think that it is not applicable to LMC, as we are not running LD with a changing target distribution, and thus there is no action $\cal A$. However, this approach based on action may be applied to the study of diffusion model and classifier-free guidance, which we will study in the future.
> > >
> > > **Questions 2**: *What are the estimates of the action parameter $\cal A$ for a more general class of distributions than the mixture of Gaussians?*
> > >
> > > **Response**: The estimate of $\cal A$ for more general target distributions is hard to obtain, as the Wasserstein distance does not have close form in general. One possible way is to leverage Lem. 3, as long as we have upper bounds on the LSI/PI constant. Yet as Ex. 1 suggests, the obtained upper bound may be loose. This problem will be the focus of our future study.
> > >
> > > We sincerely appreciate the time and effort you have dedicated to providing thoughtful feedback. We hope that our response has successfully addressed your concerns.

---

> > > > ### Comment · Reviewer_WqFj · 2024-11-27
> > > >
> > > > **Weakness 1**: The condition is indeed mild. However, for the sake of completeness, the theory should include a claim justifying why this assumption holds a priori. This claim needs to be explicitly stated and supported with a proof.
> > > >
> > > > **Weakness 3**: Your explanation clarifies the notation well. Please include this discussion in the revised version for better readability and completeness.
> > > >
> > > > **Weakness 4 (Extended)**: The experiments do not compare the proposed method with existing approaches. Sampling from mixtures of distributions, such as those that can be addressed with LMC, is relatively straightforward (see Section 2.3 of [1]). Therefore, the paper should evaluate its method in more challenging scenarios than Gaussian mixtures.
> > > >
> > > > Additionally, the ALMC results depend on the action parameter, which may become very large in the case of mixture distributions. Given that Gaussian mixtures are "easy" to sample from using Langevin methods, comparing the proposed approach with standard LMC under a global LSI condition is not particularly informative. While the proposed method might remain relevant in more general cases, the behavior of the action parameter under these conditions still needs thorough investigation.
> > > >
> > > > **Weakness 5**: I understand your point, and I agree with it. What I intended to convey is that deriving convergence rates up to a constant would enhance the precision of the results.
> > > >
> > > > **Additional Comments**: The rest of my concerns were adequately addressed, and I appreciate the effort made to clarify these points. As a result, I have increased my score. However, I still believe the paper requires further improvements, particularly in its experimental validation and theoretical completeness.
> > > >
> > > > ---
> > > >
> > > > **Reference**:
> > > > [1] Dalalyan, Arnak S., and Avetik Karagulyan. "User-friendly guarantees for the Langevin Monte Carlo with inaccurate gradient." *Stochastic Processes and their Applications* 129.12 (2019): 5278-5311.
> > > >
> > > > ---

---

> > > > > ### Author Response · Authors · 2024-11-28
> > > > >
> > > > > We would like to sincerely thank you for raising the rating and we appreciate your kind suggestions for improvement.
> > > > >
> > > > > Several slight modifications have been made in the revised manuscript: we have included a new Lem. 4 in App. A, which is a formal statement of a sufficient condition for absolute continuity in the rebuttal, and have included the discussion on smoothness as a footnote on page 3.
> > > > >
> > > > > For weakness 5, the complexity bound can actually written as $O\left(\frac{d\beta^2{\cal A}^2}{\varepsilon^6}\right)+\widetilde O(1)$, where $O\left(\frac{d\beta^2{\cal A}^2}{\varepsilon^6}\right)$ is the dominant complexity for running ALMC, and the $\widetilde O(1)$ part is for sampling from $\pi_0$. It is actually $O(1)$ when $\eta _0=0$, and $O\left(1\vee\log\frac{\eta _0\beta R^2}{d^2}\right)$ otherwise (see Lem. 5). As the $\log$ part is in general small, this complexity can be absorbed in $O\left(\frac{d\beta^2{\cal A}^2}{\varepsilon^6}\right)$.
> > > > >
> > > > > Further experimental validation and theoretical completeness on more general non-log-concave distributions will be the focus of our future study. Thank you again for your valuable comments.

---

### Official Review · Reviewer_KRFS · 2024-11-01

**Soundness:** 4
**Presentation:** 4
**Contribution:** 3
**Rating:** 8
**Confidence:** 3

**Summary:**

The paper proves new bounds on efficacy of Annealed Langevin MC by creating an SDE which approximates the target and then analyzing the discretization error introduced by simulating it on a finite machine

**Strengths:**

Computational bounds are generic- a smoothness constraint for target distribution, rather than isoperimetry assumptions.

Explanations are clear and well-illustrated, suiting a broad generalist audience by mixing intuition-building and commentary.

The direct impact of this paper is unclear, but the methods it introduces are interesting contributions

**Weaknesses:**

The paper shares the weakness of most such oracle inequalities, which is that they reassure us of convergence of an algorithm under assumptions that are hard to verify on a particular problem

Note that I am not expert in all the components of the authors' proof, and have taken some steps on faith.

**Questions:**

What does $\beta$-smoothness imply about the target distribution? Can we know this in general, especially where the density arises from a posterior? It seems like this quantity would be hard to calculate for even tractable likelihoods, let alone general posterior densities.

---

> ### Author Response · Authors · 2024-11-21
> **Response to reviewer KRFS**
>
> Thank you for your appreciation of our work and instructive feedbacks.
>
> **Weakness**: The paper shares the weakness of most such oracle inequalities, which is that they reassure us of convergence of an algorithm under assumptions that are hard to verify on a particular problem.
>
> **Response**: We agree that verifying these assumptions can be challenging in specific applications. However, our goal in establishing these conditions is to provide a broadly applicable theoretical framework that characterizes the behavior of annealed sampling in a general setting. These assumptions are intended to capture typical scenarios that arise in sampling from complex, multimodal distributions, allowing our results to extend to a wide range of problems.
>
> **Question**:
> 1. *What does $\beta$-smoothness imply about the target distribution?*
>
>     **Response**: Consider a target distribution $\pi\propto{\rm e}^{-V}$, where the potential $V$ is $\beta$-smooth. By definition, it means $-\beta I\preceq\nabla^2V\preceq\beta I$, which is equivalent to
>
>     \begin{cases}V(y)\le V(x)+\left\langle\nabla V(x),y-x\right\rangle+\frac\beta2\|y-x\|^2\\\\ V(y)\ge V(x)+\left\langle\nabla V(x),y-x\right\rangle-\frac\beta2\|y-x\|^2\end{cases}
>
>     Intuitively, this means the potential can be upper and lower bounded by quadratic functions at every point, meaning that it does not have regions of extremely high or low curvature. This assumption on the regularity of the target distribution aids in controlling the convergence of ALMC and in establishing the oracle complexity bounds.
>
> 2. *Can we know this in general, especially where the density arises from a posterior? It seems like this quantity would be hard to calculate for even tractable likelihoods, let alone general posterior densities.*
>
>     **Response**: Our paper focuses on the *theoretical* analysis of sampling, where we assume the potential is smooth to simplify the problem. In practice, the smoothness of the target distribution's potential requires case-by-case verification, and is calculable in some target distributions.
>
>     For instance, consider the Bayesian logistic regression problem: the parameter has prior distribution $\beta\sim{\cal N}(0,\eta^2I)$, and the likelihood is $Y|X,\beta\sim{\rm Bernoulli}(\sigma(\beta^{\rm T}X))$, where $\sigma(t)=\frac{1}{1+{\rm e}^{-t}}$ is the sigmoid function. Then the posterior distribution given paired data points ${\cal D}=\{(x_i,y_i)\}_{i=1,2,...,N}$ is
>     $$p(\beta|{\cal D})\propto\exp\left(-\frac{\|\beta\|^2}{2\eta^2}-\sum_i\beta^{\rm T}x_i(1-y_i)-\sum_i\log(1+{\rm e}^{-\beta^{\rm T}x_i})\right)=:\exp(-V(\beta)).$$
>     It is easy to verify that
>     $$\nabla^2V(\beta)=\frac{1}{\eta^2}I-\sum_ix_ix_i^{\rm T}\frac{{\rm e}^{\beta^{\rm T}x_i}}{(1+{\rm e}^{\beta^{\rm T}x_i})^2}.$$
>     So for any unit vector $u$, $u^{\rm T}\nabla^2V(\beta)u=\frac{1}{\eta^2}-\sum_i(u^{\rm T}x_i)^2\frac{{\rm e}^{\beta^{\rm T}x_i}}{(1+{\rm e}^{\beta^{\rm T}x_i})^2}$, which implies $u^{\rm T}\nabla^2V(\beta)u\le\frac{1}{\eta^2}$ and $u^{\rm T}\nabla^2V(\beta)u\ge\frac{1}{\eta^2}-\sum_i(u^{\rm T}x_i)^2\ge\frac{1}{\eta^2}-\sum_i\|x_i\|^2$. Hence, $V$ is smooth with constant $\max\left(\frac{1}{\eta^2},\sum_i\|x_i\|^2-\frac{1}{\eta^2}\right)$.
>
>     We agree that verifying smoothness may be hard in some general cases, let alone computing the constant $\beta$. Nevertheless, this assumption is common in non-asymptotic analyses, as it allows us to develop a clearer understanding of the algorithm’s behavior under idealized conditions, even if such conditions may be approximations in practical scenarios.
>
> We hope that our response has successfully addressed your concerns.

---

### Official Review · Reviewer_4z2B · 2024-11-04

**Soundness:** 2
**Presentation:** 2
**Contribution:** 2
**Rating:** 6
**Confidence:** 4

**Summary:**

The paper studies the convergence of annealed Langevin algorithms for general, possibly non-log concave, target distributions. The authors propose using an argument based on Girsanov's theorem to bound the distance between a reference process and a Langevin SDE that is based on an annealing schedule of target distributions, bridging the target to a base distribution. The authors also consider a practical implementation of the continuous time SDE, based on the exponential integrator scheme.

**Strengths:**

The authors consider an interesting problem, which is actively researched. Most of the research has focused on obtaining convergence bounds for standard Langevin algorithms imposing strong assumptions on the target distribution, such as log-concavity or log-Sobolev inequalities. The approach in this paper seems original, since it differs from strategies that are typically used in the literature as explained by the authors. Moreover, it gives convergence bounds in KL for an annealed Langevin algorithm, for which few results are available, and under "weak" conditions. Overall, the paper is clear enough to follow what the authors are doing and has a cohesive story.

**Weaknesses:**

- The authors consider a specific annealing schedule and it is not clear if their results give any kind of clarity on how the schedule affects the overall complexity of the algorithm.

- In several occasions I would have appreciated clearer explanations. For instance, why considering the exponential integrator scheme instead of an Euler scheme? Why considering that specific annealing schedule? It is also not completely clear why one finds the topics in Section 2.4 at a first read. It would be beneficial to convince the reader that these mathematical concepts are needed, also giving an idea of how these are useful in what follows.

- The sketch of the proof of Theorem 2 is not very clear to me and I find it should be heavily improved. It seems the authors compute the KL between two processes, neither of which is the discretisation scheme. Then the notation in the proof in the appendix seems inconsistent, denoting by \nu^{ALMC} the measure of another process. As I motivate in the Questions section below, I struggle to see how the authors use Lemma 1 in the proof of this result. This should be clarified if the paper is to be considered for acceptance.

- the contribution is in general interesting, but I would have liked to find analyses such as some (simple) numerical simulations showing that the convergence of the algorithm is for instance not exponential in the dimension.

- Table 1 gives an overview of results available in the literature, but these are for various algorithms which are denoted by corresponding  acronyms.  This is not a very clear comparison in my opinion, and it would be nice to have a short explanation at the differences between these algorithms.

- the intuitive explanations in lines 318-323 are rather obvious and are not really clarified by the results in the paper. This should be clear to the reader to avoid confusion.

**Questions:**

- Theorem 2 uses Lemma 1, that is a consequence of Girsanov's theorem. As far as I understand it uses it to bound the KL between the SDE in Equation (6) with the "true" SDE, that is Equation (4). However, the SDE (6) clearly does not fit into Lemma 1 of the paper, since a term in the drift depends on a past state. The authors should explain their reasoning and not just refer to Lemma 1, which is not general enough.

- in Lemma 2, in what sense do the authors say the equality is "reachable"?

- in lines 373-374 the authors state that the Euler scheme is "non-optimal", without a supporting reasoning. What are their reasons for writing that?

- In Example 1, the authors say that \rho_1 is "readily sampleable". Is it then the case that \rho_0 is a simple distribution that can be sampled? Here the notation is not clear yet. Also, in point 2. of the example, are the LSI and PI constants exponentially dependent on d for the distribution \rho_t for any t? In general, this example could be made clearer.

---

> ### Author Response · Authors · 2024-11-21
> **Response to reviewer 4z2B (1)**
>
> Thank you for your careful reading of our paper and your valuable, constructive feedback.
>
> **Weakness 1**: *The authors consider a specific annealing schedule and it is not clear if their results give any kind of clarity on how the schedule affects the overall complexity of the algorithm.*
>
> **Response**: The continuous-time analysis we presented in the paper is general and applies to any AC interpolation. In practice, the affect of annealing schedule to the complexity is a crucial question.
>
> In Lem. 3(1), we showed that for fixed $\rho _0$ and $\rho _1$, the curve $\rho$ that has the minimial action ${\cal A}=\int _0^1|\dot\rho| _t^2{\rm d}t$ is the constant-speed Wasserstein geodesic. However, this does not have close form except in some trivial cases such as Gaussian distribution.
>
> To realize constant-speed Wasserstein geodesic, one may do time-reparameterization on the Wasserstein gradient flow. For example, consider the OU process ${\rm d}Y _t=-Y _t{\rm d}t+\sqrt2{\rm d}B _t$, $Y _0\sim\pi$ (target distribution) and $Y _t\sim p _t$. The OU process is the Wasserstein gradient flow of the functional ${\rm KL}(\cdot\|{\cal N}(0,I))$, and after time-reparameterization of the curve $(p _t) _{t\in[0,\infty)}$, one may obtain a curve close to the constant-speed Wasserstein geodesic between $\pi$ and ${\cal N}(0,I)$. This curve is widely used in diffusion model, but we do not have closed form of the scores $\nabla\log p _t$ in general cases. As our goal is to obtain the complexity of learning-free sampling algorithms, we do not study this approach in the paper. Instead, we focus on the curve $\pi _\theta\propto\exp\left(-\eta(\theta)V-\frac{\lambda(\theta)}{2}\|{\cdot}\|^2\right),~\theta\in[0,1]$ in this paper. The effect of general annealing schedule to the sampling complexity will be the focus of our future study.
>
> **Weakness 2**:
>
> 1. *Why considering the exponential integrator scheme instead of an Euler scheme?*
>
>     **Response**: Since exponential integrator scheme reduces discretization error compared with Euler scheme. Recall that the SDE of ALD is
>     $${\rm d}X _t=\left(-\eta\left(\frac{t}{T}\right)\nabla V(X _{t})-\lambda\left(\frac{t}{T}\right)X _t\right){\rm d}t+\sqrt{2}{\rm d}B _t,~t\in[0,T].$$
>     Assume the discretization time points are $0=T _0<T _1<...<T _M=T$. On $[T _{\ell-1},T _\ell]$, the Euler discretization scheme regards the linear part in the drift term $-\lambda\left(\frac{t}{T}\right)X _t$ as unchanged, but actually the integral of $-\lambda\left(\frac{t}{T}\right)X _t$ on $[T _{\ell-1},T _\ell]$ can be computed in closed form, and by doing this, the exponential integrator scheme reduces discretization error. This is also a standard approach in accelerating the sampling of diffusion models.
>
> 2. *Why considering that specific annealing schedule?*
>
>     **Response**: This specific family of interpolation curves is a popular choice in practice (see the last paragraph of section 5.1 for discussion). Moreover, another reason for choosing this specific family of interpolation curves is that the score functions $\nabla\log\pi _\theta$ have closed form, and therefore, unlike many other curves such as the Gaussian convolution curve $\pi _\theta=\pi*{\cal N}(0,\sigma^2(\theta)I)$, we do not need further training for approximating the score.
>
> 3. *It is also not completely clear why one finds the topics in Section 2.4 at a first read.*
>
>     **Response**: Section 2.4 provides some essential technical tools in optimal transport for establishing our results. For instance, Lem. 2 is used in the proof of Thms. 1 and 2 to minimize the integral of squared $L^2$ norms, and the action will turn out to be a crucial property of the annealing curve (see, e.g., Ex. 2). Lem. 3 is on the relationship between action and isoperimetric inequalities, which sets this approach apart from the traditional analysis based on LSI or PI. We have added several linking sentences to inform the readers of the purpose of introducing these concepts and properties.

---

> > ### Author Response · Authors · 2024-11-21
> > **Response to reviewer 4z2B (2)**
> >
> > **Weakness 3**: The sketch of the proof of Theorem 2.
> >
> > 1. *The authors compute the KL between two processes, neither of which is the discretisation scheme.*
> >
> >     **Response**: Please refer to Figure 1 for the illustration. The two processes are: the reference SDE (Eq. 4) with path measure $\mathbb{P}$, which is a continuous-time process, and the ALMC SDE (Eq. 6) with path measure $\mathbb{Q}$, which is a discretized process, although we write it in the form of an SDE. The explicit update form for practical use is detailed in Alg. 1.
> >
> > 2. *Then the notation in the proof in the appendix seems inconsistent, denoting by $\nu^{\rm ALMC}$ the measure of another process.*
> >
> >     **Response**: $\nu^{\rm ALMC}$ is defined as the distribution of $X _T$ in the ALMC SDE (Eq. 6), i.e., the marginal distribution of the path measure $\mathbb{Q}$ at final time $T$.
> >
> > 3. *I struggle to see how the authors use Lemma 1 in the proof of this result.*
> >
> >     **Response**: Please see the response of Question 1.
> >
> > **Weakness 4**: *I would have liked to find analyses such as some (simple) numerical simulations showing that the convergence of the algorithm is for instance not exponential in the dimension.*
> >
> > **Response**: Please see Sec. 6 for the experimental results, where we studied the complexity w.r.t. $r$, the same norm of the means. As discussed in the respoonse of question 4 below, in this case we show an improved bound on $r$ instead of $d$, which we have corrected in the paper.
> >
> > **Weakness 5**: *Table 1 [...] It would be nice to have a short explanation at the differences between these algorithms.*
> >
> > **Response**: Thanks for your advice. We have modified the paper and added the discription of the algorithms in Sec. 1, before the table.
> >
> > **Weakness 6**: *The intuitive explanations in lines 318-323 are rather obvious and are not really clarified by the results in the paper. This should be clear to the reader to avoid confusion.*
> >
> > **Response**: In Thm. 1, we have shown that for any given AC curve of probability measures $(\pi _\theta) _{\theta\in[0,1]}$ and a sufficiently long time $T$, by reparametrizing the curve and running ALD from time $0$ to $T$, then the KL divergence from the target distribution $\pi=\pi _1$ to the obtained distribution $\nu^{\rm ALD}$ at time $T$ is $\le\frac{\cal A}{4T}$. This shows that with longer $T$, we are moving more slowly and the discrepancy between the target and the obtained distribution diminishes. This suggests the intuitive explanations in this paragraph.
> >
> > **Question 1**: *The SDE (6) clearly does not fit into Lemma 1 of the paper, since a term in the drift depends on a past state.*
> >
> > **Response**: Thank you for the advice. The Girsanov theorem also works for the case where the process $Y$ is *adaptive* (not necessarily Markov). More specifically, consider a probability space $(\Omega,{\cal F},({\cal F} _t) _{t\ge0},\mathbb{P})$ on which $W=(W _t,{\cal F} _t) _{t\ge0}$ is a standard $d$-dimensional Brownian motion. We say that $Y=(Y _t,{\cal F} _t) _{t\ge0}$ is adaptive w.r.t. the filtration ${\cal F} _t$ if $Y _t$ is a ${\cal F} _t$-measurable random vector for all $t\ge0$. For instance, in our paper, the process $Y$ may take the differential form ${\rm d}Y _t=\color{red}{b _t(Y _{t _-})}{\rm d}t+\sqrt{2}{\rm d}B _t$, where $t\mapsto t _-$ is a piecewise constant function with $t _-\le t$, corresponding to the time point of discretization. In this case, the Girsanov theorem has the form
> > $$\operatorname{KL}(\mathbb{P}^X\|\mathbb{P}^Y)=\operatorname{KL}(\mu\|\nu)+\frac{1}{4}\mathbb{E} _{X\sim\mathbb{P}^X}\int _0^T\|a _t(X _t)-\color{red}{b _t(X _{t _-})}\|^2{\rm d}t.$$
> > For more details, please check Thm. 3.2.6 of the book [Log-concave Sampling](https://chewisinho.github.io/main.pdf). We have modified the paper accordingly.
> >
> > **Question 2**: *In Lemma 2, in what sense do the authors say the equality is "reachable"?*
> >
> > **Response**: This means for any AC curve of probability measures $(\rho _t) _{t\in[a,b]}$, there exists a vector field $(v _t^*) _{t\in[a,b]}$ that generates $(\rho _t) _{t\in[a,b]}$, such that $|\dot\rho| _t=\|v _t^*\| _{L^2(\rho _t)}$ for Lebesgue-a.e. $t\in[a,b]$. We have modified the statement of this lemma accordingly, and hope that this would be clearer to the reader.
> >
> > **Question 3**: *In lines 373-374 the authors state that the Euler scheme is "non-optimal", without a supporting reasoning. What are their reasons for writing that?*
> >
> > **Response**: Here, by "non-optimal", we are expressing that there exists a better discretization scheme of the SDE 3, in the sense that the discretization error can be reduced. The reason for non-optimality is explained in the paragraph after it, and the key reason is that the integral of the linear term can be computed in closed form.

---

> > > ### Author Response · Authors · 2024-11-21
> > > **Response to reviewer 4z2B (3)**
> > >
> > > **Question 4**:
> > >
> > > 1. *In Example 1, the authors say that $\rho _1$ is "readily sampleable". Is it then the case that $\rho _0$ is a simple distribution that can be sampled? Here the notation is not clear yet.*
> > >
> > >     **Response**: In this example, $\rho _0$ can be any general target distribution (which may not be easy to sample from), and $\rho _1$ is the convolution of $\rho _0$ and a Gaussian noise with high variance, which is easier to sample from (i.e., readily sampleable). As the noise level $2S$ is large enough, samples from ${\cal N}(0,2SI)$ can serve as approximate samples from $\rho _0$. Note that it is different from the later setting where $\pi _0$ is a distribution easy to sample from, and $\pi _1$ is the target.
> > >
> > > 2. *Are the LSI and PI constants exponentially dependent on $d$ for the distribution $\rho _t$ for any $t$?*
> > >
> > >     **Response**: Sorry about the mistake we made here: after carefully checking the reference, we found that the dimensional dependence may not be exponential; however, the dependence of the maximum distance between the modes should be exponential in general. We have modified Ex. 1 and its proof accordingly.
> > >
> > >     To illustrate this, consider a simpler case where there are only 2 Gaussian components: $\rho=\frac12{\cal N}(0,\sigma^2I)+\frac12{\cal N}(y,\sigma^2I)$. [Schlichting, 2019](https://www.mdpi.com/1099-4300/21/1/89) has shown that $C _{\rm PI}(\rho)\le C _{\rm LSI}(\rho)\le\frac{\sigma^2}{2}\left({\rm e}^{\|y\|^2/\sigma^2}+3\right)$, while [Dong & Tong, 2022](https://www.sciencedirect.com/science/article/pii/S0304414922001314) has shown that when $\|y\|\gtrsim\sigma\sqrt d$, $C _{\rm LSI}(\rho)\ge C _{\rm PI}(\rho)\gtrsim\frac{\sigma^4}{\|y\|^2}\exp\left(\Omega\left(\frac{\|y\|^2}{\sigma^2}\right)\right)$. Hence we conclude that the LSI/PI constants of $\rho _t$ would depend on the maximum distance between the modes exponentially in this setting when $t$ is close to $0$, but the dimensional dependence might not be exponential.
> > >
> > > We sincerely appreciate the time and effort you have dedicated to providing thoughtful feedback. We hope these our response aligns with your expectations.

---

> ### Comment · Reviewer_4z2B · 2024-11-26
>
> I thank the authors for their clarifications and modifications. I have changed my grade to 6.

---

> > ### Author Response · Authors · 2024-11-28
> >
> > Dear reviewer, thank you for raising the score! We would like to know if there‘s any additional concern that you still have regarding this work, and we would be eager to address them thoroughly.

---

### Author Response · Authors · 2024-11-21
**General response**

We would like to extend our sincere gratitude to all the reviewers for their insightful comments and valuable feedback. We are pleased that the reviewers recognized the importance of the annealed sampling problem addressed in the paper (reviewer 4z2B), appreciated the novelty of our technical contributions (reviewers 4z2B, KRFS, WqFj, WbhJ), and commended the clarity of our presentation (reviewers 4z2B, KRFS, WbhJ). Our analysis introduces a novel framework for understanding annealing algorithms through the action of the interpolation trajectory, setting our work apart from existing bounds by avoiding dependence on isoperimetric inequalities. We believe this framework is going to significantly advance the understanding of annealed sampling algorithms and pave the way for future research directions, including the design of annealing schedules.

While *all* reviewers agreed on the novelty of our approach, two primary concerns were raised:

1. The need to improving the writing in several parts of the paper to enhance both accuracy and readability for a general audience.
2. The lack of simple numerical results to validate the theoretical findings presented in the paper.

In our revised manuscript, we have made substantial improvements to address these concerns, providing clarifications for all raised questions. Notable changes in the main paper are marked $\color{red}{\rm red}$ and enumerated below:

- We have added descriptions of the algorithms compared in Tab. 1.
- The statement of Girsanov's theorem (Lem. 1) has been modified to include cases where the drift terms in the SDE depend on the past state.
- We have clarified the statement "the equality is reachable" in Lem. 2. and expanded the discussion on the motivation for introducing OT concepts in Sec. 2.4.
- An error in Ex. 1 related to the dimensional dependence of LSI/PI constants has been corrected, along with adjustments to the corresponding proof in App. A.
- We added a footnote in the proof of Thm. 1 for rigor.
- In Sec. 6, we added a numerical experiment to verify the findings in Ex.2, and showed that the dependence of the number of iterations required to achieve a certain accuracy in KL divergence depends polynomially on $r$.
- Finally, we slightly modified the layout of the paper to fit the additional content.

We have also addressed all concerns in the individual responses below. We hope that the changes we have made reflect our commitment to improving the quality of our work and that the revised manuscript meets your expectations.

---

### Meta-Review · Area_Chair_2bu3 · 2024-12-22

**Metareview:**

This paper provides a non-asymptotic analysis of annealed MCMC. Authors establish an oracle complexity of a simple variant of annealed Langevin Monte Carlo and provide convergence guarantees for this algorithm to the target in KL divergence. Target is assumed to have a , smooth potential and the rate is in terms of action of a curve of measures that interpolates the target, which is an intriguing parameter.


This paper was reviewed by four reviewers the following Scores/Confidence: 8/4, 8/3, 6/4, 6/4. I think the paper is studying an interesting topic and the results are relevant to ICLR community. The following concerns were brought up by the reviewers:

- It is not clear if the rate is tight (most likely it is not). As is, the result seems to be another upper bound on sampling from a class of targets.

- Assumptions are not discussed in detail and compared with standard ones in the literature. This should be improved.

- Even though this is a theoretical paper, a detailed experimentation section would make the paper much stronger.

- minor typos were pointed by the reviewers. These should be carefully addressed. One reviewer requested detailed explanation of the integrator.


Authors should carefully go over reviewers' suggestions and address any remaining concerns in their final revision. Based on the reviewers' suggestion, as well as my own assessment of the paper, I recommend including this paper to the ICLR 2025 program.

**Additional Comments On Reviewer Discussion:**

Reviewer questions are thoroughly answered by the authors. The revision provides colour-coded updates.

---

### Decision · Program_Chairs · 2025-01-22

Accept (Poster)